# Accelerated brain ageing during the COVID-19 pandemic

Ali-Reza Mohammadi-Nejad ®[1,2,3], Martin Craig ®[2], Eleanor F. Cox[1,4], Xin Chen ®[5], R. Gisli Jenkins[6,7,8], Susan Francis[1,4], Stamatios N. Sotiropoulos ®[1,2,3,9] ✉ & Dorothee P. Auer ®[1,2,3,9] ✉

The impact of SARS-CoV-2 and the COVID-19 pandemic on brain health is recognised, yet specific effects remain understudied. We investigate the pandemic's impact on brain ageing using longitudinal neuroimaging data from the UK Biobank. Brain age prediction models are trained from hundreds of multimodal imaging features using a cohort of 15,334 healthy participants. These models are then applied to an independent cohort of 996 healthy participants with two magnetic resonance imaging scans: either both collected before the pandemic (Control groups), or one before and one after the pandemic onset (Pandemic group). Our findings reveal that, even with initially matched brain age gaps (predicted brain age vs. chronological age) and matched for a range of health markers, the pandemic significantly accelerates brain ageing. The Pandemic group shows on average 5.5-month higher deviation of brain age gap at the second time point compared with controls. Accelerated brain ageing is more pronounced in males and those from deprived socio-demographic backgrounds and these deviations exist regardless of SARS-CoV-2 infection. However, accelerated brain ageing correlates with reduced cognitive performance only in COVID-infected participants. Our study highlights the pandemic's significant impact on brain health, beyond direct infection effects, emphasising the need to consider broader social and health inequalities.

Apart from the well-documented respiratory and systemic manifestations of SARS-CoV-2, compelling evidence highlights its neurotropic nature, showing high rates of persistent respiratory symptoms, fatigue, depression, post-traumatic stress disorder, and cognitive impairment in COVID-19 survivors[1]. Emerging research has revealed potential associations between COVID-19, cognitive decline, brain changes[2], and the molecular signatures of brain ageing[3]. Significant psychological distress and mental health issues were also reported during the early pandemic phases, especially among younger and vulnerable individuals[4]. Conversely, recent reviews suggest variable reductions in mental health service use[5] and, across 134 cohort studies, no overall rise in mental health conditions was found in the general population, with minimal increase in depression symptoms and small negative effects in women[6]. Understanding the pandemic's effects on

[1]National Institute for Health Research (NIHR) Nottingham Biomedical Research Centre, Queen's Medical Centre, Nottingham, United Kingdom. [2]Sir Peter Mansfield Imaging Centre, School of Medicine, University of Nottingham, Nottingham, United Kingdom. [3]Mental Health and Clinical Neurosciences, School of Medicine, University of Nottingham, Nottingham, United Kingdom. [4]Sir Peter Mansfield Imaging Centre, School of Physics and Astronomy, University of Nottingham, Nottingham, United Kingdom. [5]School of Computer Science, University of Nottingham, Nottingham, United Kingdom. [6]National Heart and Lung Institute, Imperial College London, London, United Kingdom. [7]National Institute for Health Research (NIHR) Imperial Biomedical Research Centre, Imperial College London, London, United Kingdom. [8]Department of Interstitial Lung Disease, Royal Brompton and Harefield Hospital, Guys and St Thomas' NHS Foundation Trust, London, United Kingdom. [9]These authors contributed equally: Stamatios N. Sotiropoulos, Dorothee P. Auer. ✉e-mail: stamatios.sotiropoulos@nottingham.ac.uk; dorothee.auer@nottingham.ac.uk

brain health, considering infection status and socio-demographic factors, is crucial for addressing its long-term health consequences and broader public health implications.

The neuroinvasion of SARS-CoV-2 is well established[7], with virus persistence detected up to 230 days post-infection[8]. Central nervous system manifestations have been linked to direct neuroinvasion, vascular damage, and immune responses[9]. Recent studies suggest that COVID-19 may accelerate neurodegenerative processes or contribute to age-related cognitive impairments[10]. Longitudinal assessments indicate a higher risk of cognitive decline among COVID-19 survivors compared with controls[11]. Serial brain magnetic resonance imaging (MRI) analyses have demonstrated widespread structural brain changes, including reductions in both grey matter (GM) thickness and white matter (WM) integrity, potentially due to neurodegeneration, neuroinflammation, or sensory deprivation[2]. Beyond direct infection effects, the pandemic may have independently influenced brain ageing due to psychosocial stressors, social disruptions, and lifestyle changes, particularly among vulnerable groups such as older adults[12] and individuals experiencing economic hardship.

While indirect evidence suggests that COVID-19 infection may accelerate brain ageing, comprehensive studies examining the pandemic's broader impact on brain health remain limited. Advanced neuroimaging and machine learning approaches have enabled the development of brain age prediction models, which estimate deviations from typical ageing trajectories—captured as the brain age gap (BAG), defined as the difference between estimated brain age and chronological age. Seminal works[13–15] laid the foundation for these models, refined with large-scale datasets, multi-modal imaging[16,17], extensions to brain tissue-specific models to delineate different aspects of brain ageing[18], and proven associations with mortality. Utilising these methodologies, we estimated brain age and investigated the impact of COVID-19 and the pandemic on brain age using longitudinal neuroimaging data.

We hypothesise that COVID-19 infection and the pandemic accelerated brain ageing. To test this, we utilised serial neuroimaging data from the UK Biobank (UKBB) study[19]. We trained a model using multi-modal imaging-derived phenotypes (IDPs) to predict individual BAG. The trained model was then applied to unseen participants with two brain scans, one before and one after the pandemic (Pandemic group) or both scans before the pandemic (Control group). We further assessed the impact of COVID-19 infection within the Pandemic group and explored putative moderating factors on brain ageing, such as sex and deprivation indices, and the interrelation with cognitive decline.

## Results

A brain age prediction model[16] (Fig. 1a) was trained on MRI scans collected pre-March 2020 from 15,334 healthy middle-aged and older UKBB[19] participants ('training set': 8407 female; age [mean ± SD]: 62.6 ± 7.6 years). From the full neuroimaging dataset of > 42,000 individuals, only participants classified as healthy, with no history of chronic disorders (e.g., heart disease, diabetes, dementia, kidney disease, major depression – see full list of exclusions in Supplementary Table 1 as in refs. 18,20) were included in this training set. This minimised the potential confounding effects of disease and comorbidities on brain age predictions. Hundreds of multi-modal IDPs were extracted[21] and used as regressors in the model after PCA-based dimensionality reduction (Fig. 1b). As COVID-19 may affect differently WM and GM[2,18,22,23] and susceptibility to neurological diseases can vary across sexes[24], separate models were trained based on GM and WM features, and for males and females[18].

These models were then applied to our unseen study cohort with two MRIs, comprising 996 healthy participants (552 female; age: 58.8 ± 6.2 years; mean inter-scan intervals (ΔT) of ~ 34 months – Supplementary Fig. 1), where participants with major chronic conditions before both scans were excluded to maintain consistency in health status across all subjects. The study cohort included the Pandemic group (G1: $N = 432$, 255 female) with one brain scan before and one after the pandemic, and the Control group (G2: $N = 564$, 297 female) with both scans before the pandemic. The groups were adjusted to be matched for age, sex, and other health markers (see Supplementary Table 2), and only participants with a minimum inter-scan interval of 2 years, who did not develop an interim health condition, were considered[25] (Supplementary Fig. 1d). Using the trained models, brain age was estimated for both time points of each participant. The difference between estimated brain age and chronological age (BAG) was then obtained at both time points, and the rate of change in BAG was calculated and normalised for the inter-scan intervals as $R_{BAG} = (\Delta BAG/\Delta T)$.

### Performance of brain age prediction models

Scatter plots in Fig. 1c depict the relationship between chronological and predicted brain age for each brain tissue type and sex (males shown in Supplementary Fig. 2). We employed an unbiased estimation approach for brain age[16], ensuring BAG is orthogonal to chronological age. All models demonstrated relatively similar prediction accuracy, with Pearson's r ranging from 0.905 (WM female model, $p < 0.0001$, 95% CI = 0.901–0.909; Mean Absolute Error (MAE) = 2.90 years) to 0.894 (WM male model, $p < 0.0001$, 95% CI = 0.890–0.899; MAE = 3.09 years), indicating that neuroimaging features captured a large proportion of chronological age variance, consistent with previous methodologically rigorous studies[18,26].

We further confirmed that our model's estimated brain age was unbiased towards the group mean[27], and that participants' age distribution was Gaussian[16]. For the remainder of this paper, unless otherwise stated, we aggregated the predicted brain age gap for male and female models across different brain tissue types and participant groups separately. Figure 1d shows that there was also no significant correlation between the estimated BAG and chronological age when applying the trained model to unseen data (Pearson's $r < 0.001$). As expected, we found very high correlations between predicted brain ages of participants at the two time points (Fig. 1e, Pearson's $r > 0.96$, FDR-corrected $p < 4.0e-234$), demonstrating high scan-rescan model reproducibility. The intraclass correlation coefficient (ICC) further supported this reproducibility, with an ICC of 0.981 (95% CI: 0.977–0.985) for the Pandemic group and 0.983 (95% CI: 0.980–0.985) for the No Pandemic group, indicating stability of estimated brain ages over time. In addition, partial correlation analysis, controlling for chronological age at each scan, yielded high partial correlations (Pandemic group: $r = 0.86$, 95% CI = [0.83–0.88], FDR-corrected $p = 6.3e-120$; No Pandemic group: $r = 0.88$, 95% CI = [0.87–0.90], FDR-corrected $p = 6.5e-307$). These results suggest that the reproducibility of brain age estimates reflects individual brain health properties.

No differences in mean predicted BAG was found between the training and unseen cohort's first MRI data (Mann-Whitney two-sample t-test, GM: FDR-corrected $p = 0.44$, WM: FDR-corrected $p = 0.99$), demonstrating the model's generalisability (Fig. 1f). Importantly, the estimated BAG for the first scan for the Pandemic group was not significantly different from the corresponding BAG for the Control (No Pandemic) group (GM: FDR-corrected $p = 0.23$, WM: FDR-corrected $p = 0.28$), confirming that our matching (Supplementary Table 2) effectively achieved comparable baseline BAGs.

### Accelerated Brain ageing is associated with the COVID-19 pandemic, regardless of infection

Although BAGs were not statistically different between the Pandemic and Control groups at the first time point, the pandemic's effect on brain ageing became evident with the second scan. Figure 2 presents the rates of change in BAG between the two scans ($R_{BAG}$). The Pandemic group displayed significantly higher $R_{BAG}$ compared with the Controls (GM Cohen's $d = 0.606$, WM Cohen's $d = 0.697$; FDR-corrected $p < 0.0001$), indicating accelerated brain ageing.

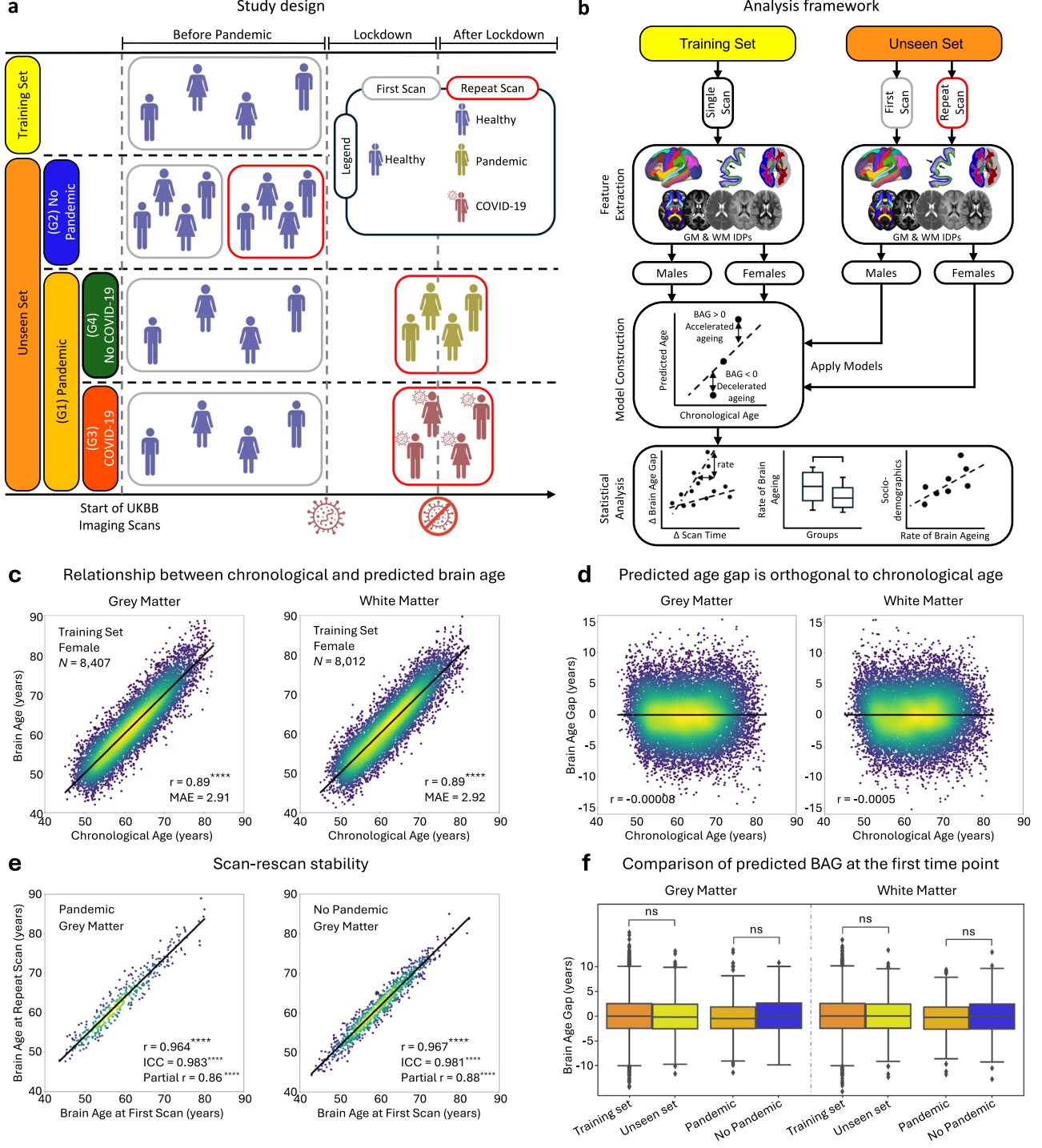

**Fig. 1 | Study design, analysis framework, and accuracy assessment of brain age prediction models. a** A brain age prediction model was trained using 20-fold cross-validation on healthy participants with a single pre-pandemic scan (training set). The model was applied to an unseen set comprising the Pandemic group (G1) and the No Pandemic group (G2). G1 was further subdivided into Pandemic–COVID-19 (G3) and Pandemic–No COVID-19 (G4). **b** Imaging-derived phenotypes (IDPs) were extracted from grey matter (GM) and white matter (WM) across scan times. Separate prediction models were trained by tissue type and sex using pre-pandemic data, and then applied independently to scans from different time points to estimate brain age gap (BAG). Statistical analyses assessed pandemic- and infection-related effects using longitudinal data. **c** Scatter plots show predicted vs. chronological age for GM and WM models in females (males shown in Supplementary Fig. 2). The diagonal line indicates perfect prediction. '*N*' is the number of subjects used for training. Model performance was evaluated using Pearson's correlation (r) and mean absolute error (MAE), averaged across 100 repetitions.

**d** Relationship between BAG and chronological age for GM and WM models, aggregated across sexes. The black regression line indicates no age-related bias. **e** Predicted brain ages at two time points show high reproducibility in both groups (Pearson's *r* > 0.96). Intraclass correlation coefficients were 0.981 (95% CI: 0.977–0.985) for the Pandemic group and 0.983 (95% CI: 0.980–0.985) for the No Pandemic group, confirming temporal stability. Partial correlation analyses, controlling for chronological age, yielded *r* = 0.86 (95% CI: 0.83–0.88) for the Pandemic group and *r* = 0.88 (95% CI: 0.87–0.90) for the No Pandemic group. **f** Boxplots compare BAG distributions between the training set (*N* = 15,334) and unseen (first scan) set (*N* = 996), and between Pandemic (*N* = 432) and No Pandemic (*N* = 564) groups for GM and WM models. No significant differences were observed (GM: p(FDR) = 0.44, 0.23; WM: p(FDR) = 0.99, 0.28). Each scatter point represents a participant. Asterisks (****) indicate FDR-corrected *p* ≤ 0.0001; 'ns' denotes non-significant differences.

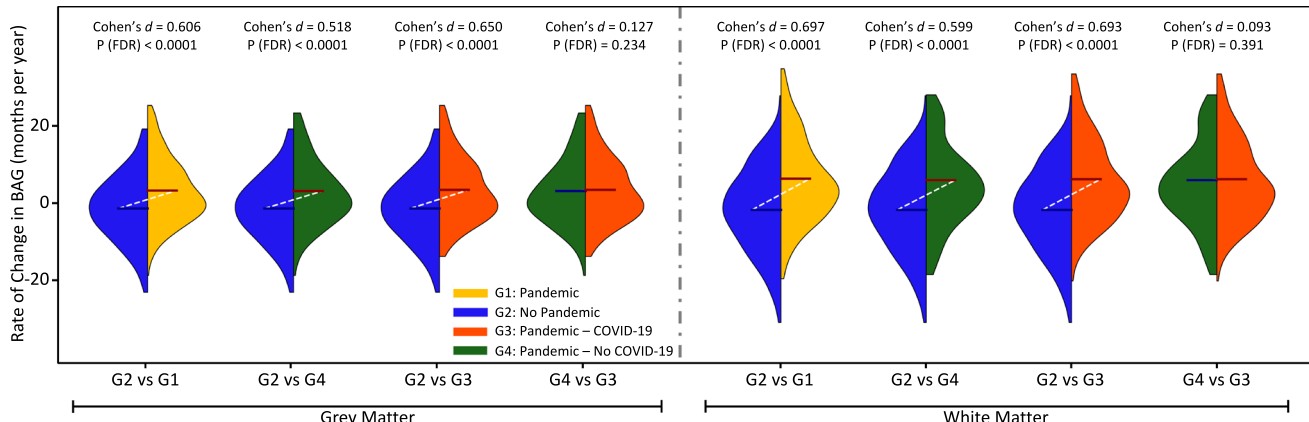

**Fig. 2 | Effect of COVID-19 and the pandemic on brain ageing.** This figure illustrates the distribution of the rate of change in brain age gap (BAG) across different brain tissue models and subject groups. The left panel corresponds to the Grey Matter (GM) model, while the right panel represents the White Matter (WM) model. Each group is displayed using coloured half-violin plots: orange for the Pandemic group (G1, $N = 432$), blue for the No Pandemic group (G2, $N = 564$), red for the Pandemic–COVID-19 group (G3, $N = 134$), and green for the Pandemic–No COVID-19 group (G4, $N = 298$). The y-axis indicates the rate of change in brain age gap in months per year. Pairwise comparisons between groups were performed using two-sample $t$ tests, with $p$-values corrected for multiple comparisons using FDR. Cohen's $d$ values, which quantify the effect size of group differences, were also calculated.

To further investigate whether SARS-CoV-2 infection specifically influenced accelerated brain ageing, the Pandemic group was subdivided into: Pandemic–COVID-19 (G3), with participants who had COVID-19 (134 participants–78 females, Supplementary Fig. 1g), and Pandemic–No COVID-19 (G4), with individuals without reported infection before the second scan (298 participants–177 females, Fig. 1a and Supplementary Fig. 1a, b). Notably, these subgroups were also adjusted to be matched to controls to ensure comparability (Supplementary Table 2). As illustrated in Fig. 2, both G3 and G4 had higher $R_{BAG}$ values than No Pandemic controls (Cohen's $d = 0.518$ (0.599) for No COVID-19 vs Controls in GM (WM) and Cohen's $d = 0.65$ (0.693) for COVID-19 vs Controls in GM (WM), respectively), with no significant difference between subgroups for either GM or WM models (Supplementary Fig. 3 shows brain age gap distributions at various time points across groups). This suggests increased positive brain age deviation (accelerated brain ageing) during the pandemic, regardless of SARS-CoV-2 infection.

**Effects of age and sex on longitudinal brain ageing (rate of change in BAG)**

While the estimated BAG was designed to be independent of chronological age at a single time point, biologically plausible longitudinal effects on $R_{BAG}$ cannot be excluded. Understanding whether brain age acceleration varies across different age groups can reveal periods of increased vulnerability and potential dependencies on infection status and tissue specificity[28–30]. To examine this, we regressed $R_{BAG}$ against the average chronological age between the two scans $t_1$ and $t_2$, calculated as $AvgAge = (Age_{t_1} + Age_{t_2})/2$, following previous studies[18,31]. Using the average age rather than individual time points in a longitudinal analysis helps mitigate potential biases and accounts for variations in scan intervals across participants[32,33].

Across all groups, a positive association was observed between average chronological age and accelerated brain ageing (Supplementary Fig. 4), suggesting that older individuals exhibited greater increases in BAG over time. This effect was strongest in the Pandemic group (Fig. 3a), where participants with a higher average age exhibited more pronounced $R_{BAG}$ acceleration compared with Controls.

In Controls, each 1-year increase in average chronological age was associated with an approximate BAG acceleration of 3 days for both GM (FDR-corrected $p = 0.0027$) and WM (FDR-corrected $p = 0.002$) models. In contrast, participants in the Pandemic group demonstrated a twofold higher rate of BAG acceleration, with each additional year of average age corresponding to 7 days in GM (FDR-corrected $p = 0.0048$) and 8 days in WM (FDR-corrected $p = 0.0005$) (Supplementary Fig. 4).

Further stratification revealed an age-related acceleration of BAG based on infection status. The strongest age-related BAG increase was observed in the Pandemic–COVID-19 subgroup (G3), where each 1-year increase in average age between the two scans was linked to a 9-day acceleration in GM (FDR-corrected $p = 0.004$) and 10 days in WM (FDR-corrected $p = 0.0069$) (Fig. 3a). In contrast, the Pandemic–No COVID-19 subgroup exhibited a slightly lower but still significant effect (6 days for GM, FDR-corrected $p = 0.007$; 8 days for WM, FDR-corrected $p = 0.0069$).

The pandemic's impact on accelerated brain ageing (higher $R_{BAG}$ compared to Controls) was evident in both male and female participants (Fig. 3b; Cohen's $d > 0.660$, FDR-corrected $p < 0.0001$). We used two-factor, two-level permutation tests (5000 permutations) to assess the interplay between the pandemic, sex, and their interactions on brain ageing. These tests confirmed the pandemic as a significant factor for $R_{BAG}$ (FDR-corrected $p = 0.002$ in both models–less than the 95% CI [0.0443–0.0564], calculated using the Wilson method[34]). In addition, sex was a significant factor in the GM model (FDR-corrected $p = 0.036$–less than the 95% CI [0.0443–0.0564]), but not in the WM model. Interestingly, a significant interaction (FDR-corrected $p = 0.008$–less than the 95% CI [0.0443–0.0564]) between sex and pandemic status was also found (for the GM model), indicating that the combination of the pandemic and being a male led to the highest $R_{BAG}$ increases (33% more in males vs. females). The interaction plots (Fig. 3b) demonstrate divergence between males and females when comparing the No Pandemic with the Pandemic group, highlighting the interaction between sex and the pandemic on GM-related brain ageing.

**Increased brain age gap rate during pandemic in deprived areas**

Besides age and sex, socio-demographic factors can influence brain health, cognitive reserve, and resilience to the detrimental effects of the pandemic[35–37]. The effects of deprivation indices (available in the UK Biobank) as drivers of poor brain health–such as health, employment, education, housing, and income–on brain ageing were examined.

The month-based clocks in Fig. 4a illustrate the extent of accelerated brain ageing among participants with varying deprivation levels, highlighting changes from before to during the pandemic. The

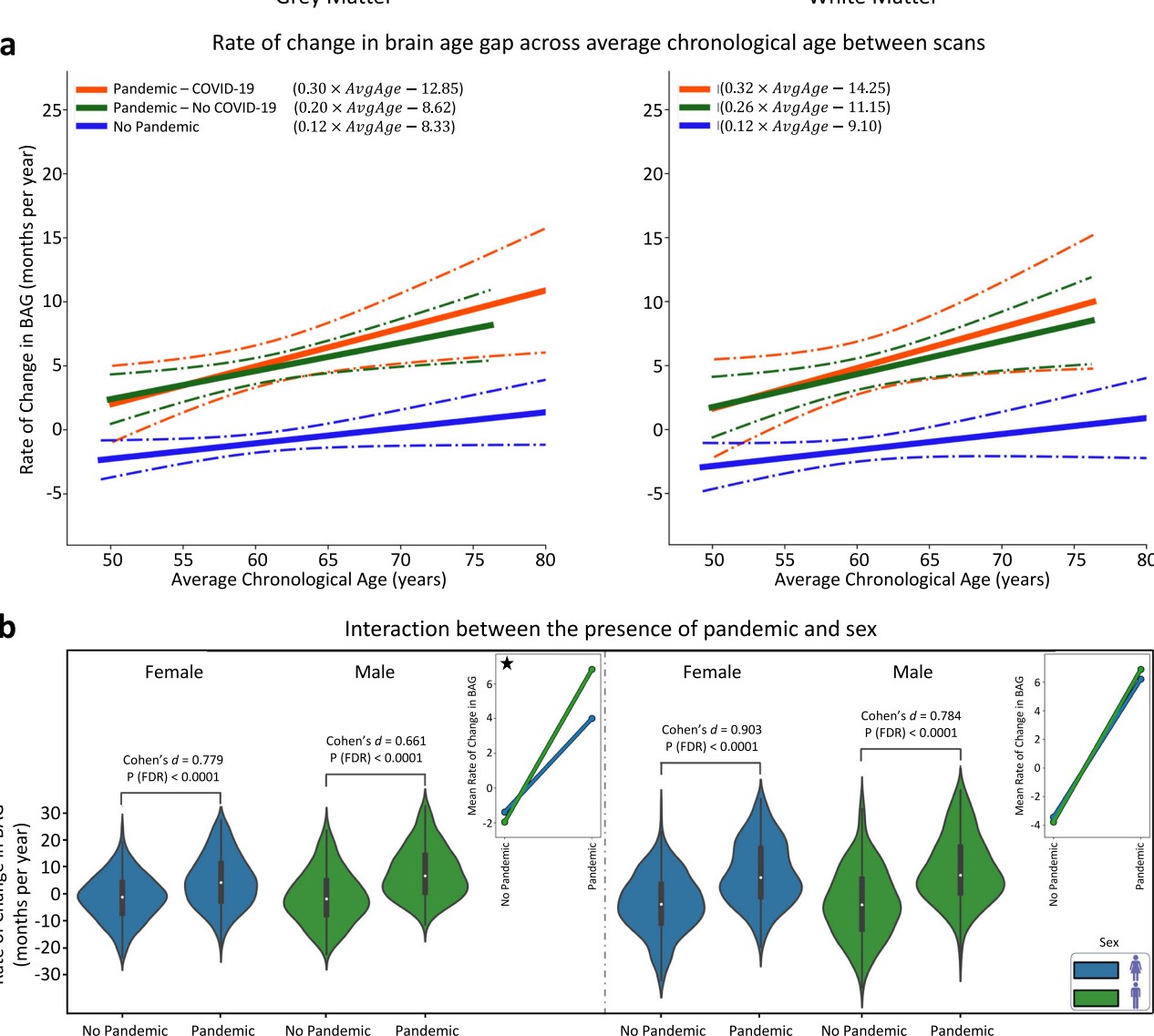

**Fig. 3 | Impact of SARS-CoV-2 infection and the COVID-19 pandemic on brain ageing, and the role of age and sex. a** Rate of change in brain age gap (BAG) is plotted against the average chronological age between two scans for the Pandemic–COVID-19, Pandemic–No COVID-19, and No Pandemic groups. Solid lines show best-fit associations; dot-dashed curves indicate 95% confidence intervals. **b** Violin plots display the distribution of the rate of change in brain age gap stratified by sex and pandemic status. For females: Pandemic group (G1), $N = 255$; No Pandemic group (G2), $N = 297$. For males: G1, $N = 177$; G2, $N = 267$. Cohen's $d$-values, representing effect sizes, are reported for each comparison, alongside the FDR-corrected $p$-values from two-sample $t$ tests between the groups. Interaction plots on the right highlight distinct patterns in grey matter (GM) and (white matter) WM between groups. Stars in the interaction plots indicate significant results, based on the FDR-corrected $p$-values of the interaction analysis determined by the two-factor, two-level permutation test. GM model results are displayed on the left and WM model results on the right in both panels.

largest increases were seen in participants with different employment scores (despite all considered participants being free from major chronic health conditions), showing a difference of about 5 months and 23 days in the GM model. This suggests an average increase of 5.8 months in $R_{BAG}$ between participants with low vs. high employment scores following exposure to the pandemic. Similarly, substantial changes were noted for low vs. high health indices (4 months and 9 days increase), and low vs. high income levels (1 month and 17 days) in the GM model. The WM model showed significant $R_{BAG}$ changes for low health index (5 months and 27 days increase), low employment index (5 months and 2 days), low education (4 months and 13 days), and low income (1 months and 8 days).

Further analysis revealed significant differences (FDR-corrected $p < 0.0001$) in brain ageing patterns between the Pandemic and No Pandemic groups across the deprivation indices (Fig. 4b–d). Generally, the increase in $R_{BAG}$ between the Pandemic and Control groups was higher for participants with high deprivation scores (low health, low education, and low employment) compared to those with low deprivation scores (high health, high education, and high employment). This was true for both GM and WM models, indicating potential interactions between the pandemic's effects and deprivation on brain ageing differences.

To further explore such interactions, we conducted nonparametric two-factor, two-level permutation tests. These tests confirmed the pandemic significantly drove the differences in predicted $R_{BAG}$ between the Pandemic and Control groups. Several deprivation indices also influenced differences between low and high deprivation, including employment (GM: FDR-corrected $p = 0.0004$; WM: FDR-

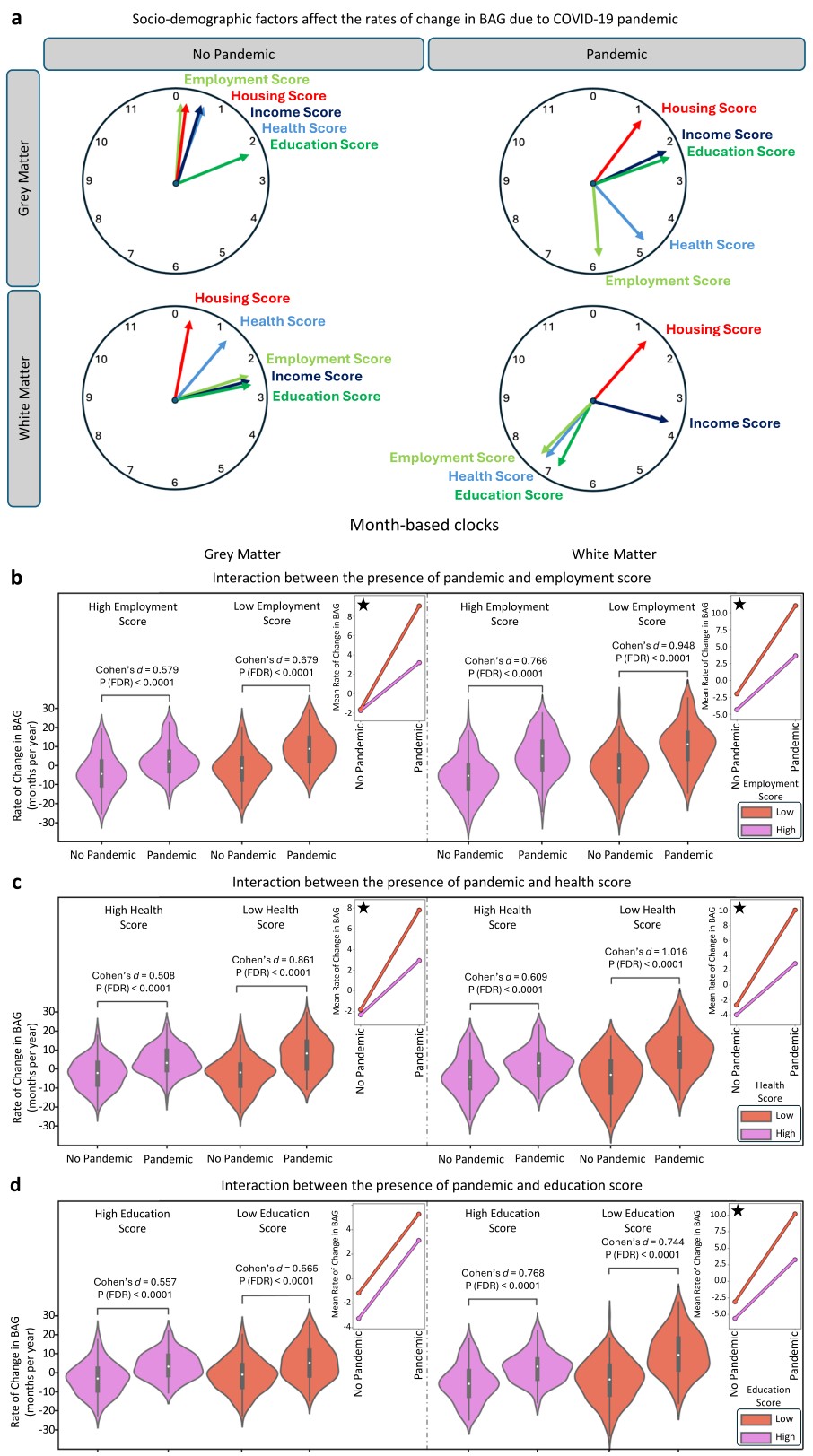

corrected $p = 0.0004$), health (GM: FDR-corrected $p = 0.0009$; WM: FDR-corrected $p = 0.0004$), education (GM: FDR-corrected $p = 0.0007$; WM: FDR-corrected $p = 0.0004$), and income score levels (GM: FDR-corrected $p = 0.0033$; WM: FDR-corrected $p = 0.0004$) (all below 95% CI [0.0443–0.0564]). Housing scores were not significant in either model.

Significant interactions between pandemic status and deprivation factors were also found (95% CI [0.0443–0.0564]). After applying FDR correction for multiple comparisons, interactions between pandemic status and employment (GM: $p = 0.0053$; WM: $p = 0.002$), health (GM: $p = 0.014$; WM: $p = 0.003$), and education scores (WM: $p = 0.0268$) on brain ageing were found to be significant. Figure 4b–d depict

**Fig. 4 | Influence of socio-demographic factors on brain ageing during the COVID-19 pandemic. a** The effects of socio-demographic factors, represented by indices of deprivation, on brain ageing in participants grouped by pandemic status. Each clock represents the difference in the mean rate of change in brain age gap (BAG) between individuals with low and high levels of specific socio-demographic factors. The clocks are presented separately for GM and WM models, with one set depicting participants in the No Pandemic group and another for participants in the Pandemic group. The socio-demographic factors studied include housing score, health score, employment score, income score, and education score. **b–d** Violin plots display the distribution of the rate of change in BAG for the Pandemic and No Pandemic groups, stratified by socio-demographic scores for **(b)** employment (No Pandemic: $N = 111$ low, $N = 129$ high; Pandemic: $N = 105$ low, $N = 102$ high), **(c)** health

(No Pandemic: $N = 110$ low, $N = 159$ high; Pandemic: $N = 111$ low, $N = 123$ high), and **(d)** education (No Pandemic: $N = 223$ low, $N = 126$ high; Pandemic: $N = 157$ low, $N = 95$ high). High and low groups are colour-coded as purple and red, respectively. Each panel includes two plots for GM (left) and WM (right) results. Cohen's $d$ effect sizes and FDR-corrected $p$-values are reported for group comparisons based on two-sample $t$ tests. Small plots on the right side of each panel depict interaction plots, suggesting the presence of interaction effects. These plots visualise how the mean rate of change in BAG deviates between the No Pandemic and Pandemic groups in both GM and WM models. Stars in the interaction plots indicate significant results based on the FDR-corrected $p$-values, calculated based on a two-factor, two-level permutation test, highlighting the interaction between the two factors.

interaction plots comparing distinct patterns in GM and WM models between the Pandemic and No Pandemic groups, highlighting socio-demographic factors' role in brain ageing during the pandemic. As sex significantly interacted with pandemic status only in the GM model (Fig. 3b), we also analysed the interplay of each deprivation index and pandemic status separately for female and male participants. Results showed that even in sex-specific models, all previous findings and interactions between pandemic and deprivation remained significant (Supplementary Fig. 5).

### Cognitive performance, accelerated brain ageing, and COVID-19 exposure

To assess the impacts of COVID-19 and the pandemic on cognitive performance related to longitudinal brain ageing, we analysed performance changes over time among individuals who completed cognitive tests at both scans. This analysis included the three groups (No Pandemic, Pandemic–COVID-19, and Pandemic–No COVID-19), focusing on the top 10 cognitive tests related to dementia risk within the UKBB[2].

Among these tests, the Pandemic–COVID-19 group showed a significantly greater decline in performance (i.e., more time to complete the test) from baseline to follow-up only for one cognitive test– the trail making test (TMT) (Fig. 5, insets). Specifically, participants in this group showed a significant increase in completion time for both TMT-A (numeric) and TMT-B (alphanumeric) compared with both the Control and Pandemic–No COVID-19 groups (Fig. 5). To account for differences in inter-scan intervals across participants, we repeated the analysis by normalising the longitudinal change in performance relative to the inter-scan interval. This adjustment did not alter the observed patterns, confirming a notable decline in cognitive function among individuals who had contracted COVID-19 (Supplementary Fig. 6).

Further analysis examined the relationship between $R_{BAG}$ and TMT-A performance using full and partial correlation analysis that excluded the effect of chronological age (Supplementary Fig. 7). A significant positive correlation was observed only in the Pandemic–COVID-19 group, suggesting that within this group, greater brain ageing changes were associated with a decline in cognitive performance. In addition, the Pandemic–COVID-19 group showed a more pronounced and non-linear decline in cognitive performance with higher $R_{BAG}$, suggesting a more prominent threshold effect for WM models and TMT-B performance. These findings suggest that while BAG increase during the pandemic was independent of COVID-19 infection, it was only associated with a decline in one cognitive test (TMT), and only in those with recorded COVID-19 (G3).

### Discussion

Using longitudinal neuroimaging data from the UKBB, we estimated individual brain age and its change rate compared to chronological ageing in two matched cohorts: one scanned before and during the COVID-19 pandemic (Supplementary Fig. 1e), and the other scanned

twice before the pandemic. We found that the COVID-19 pandemic was detrimental to brain health and induced accelerated brain ageing for GM and WM-derived models, regardless of SARS-CoV-2 infection. Accelerated brain ageing during the pandemic was more pronounced in older individuals and males based on the GM model, and in those from deprived backgrounds for both models. Cognitive performance, particularly in flexibility and processing speed tasks, declined significantly in COVID-19 infected individuals, correlating with accelerated GM ageing. Conversely, participants who experienced the pandemic without reported infection had similar age-related declines as controls, demonstrating that pandemic-related accelerated brain ageing alone was insufficient to lead to cognitive decline.

For the brain age prediction models, we relied on neuroimaging data exclusively from the UKBB[19], hence minimising potential confounds from scanner variability and protocol differences inherent to datasets pooled across different studies. At the same time, we leveraged the rich set of imaging-derived features available in thousands of UKBB participants. By training using only healthy participants without chronic disorders, we effectively constructed normative brain age models with a substantial sample size. Importantly, we employed a bias-correction step to ensure that brain age delta was independent of chronological age, mitigating potential skewing effects that could influence ageing estimates[16,26,38]. In our study, we achieved high accuracy and low MAE in brain age prediction, aligning with prior unbiased methodologies. For instance, Smith et al. [16] reported MAEs of ~2.9 years using orthogonalized brain age gap prediction models, consistent with the MAEs observed in our study (2.9–3.08 years). This unbiased approach, as further supported by refs. [26,39], consistently achieves lower MAEs compared to biased models, as highlighted by ref. [38]. These findings confirm the robustness of our model and demonstrate its comparability to state-of-the-art methods in the field.

Our findings provide valuable insight into how the COVID-19 pandemic affected brain health, demonstrating that the general pandemic effects alone, without infection, exerted a substantial detrimental effect on brain health, augmented by bio-social factors (age, health, and social inequalities) in a healthy middle-aged to older population. Notably, the extent of accelerated brain ageing over a matched pre-pandemic control group, observed in grey and white matter, was similar in both non-infected and infected sub-cohorts. This highlights the major role of pandemic-related stressors such as anxiety, social isolation, and economic, and health insecurity on brain changes that may be sufficient to explain the observed accelerated brain ageing. In other words, our findings suggest that a full bio-psycho-social model is needed to understand the negative brain health effects of COVID-19 infection during a pandemic, which previous research, such as Douaud et al.[2], has not accounted for.

Male vulnerability to brain ageing was particularly pronounced during the pandemic, consistent with prior evidence of sex differences in neurobiology. Studies (e.g.,[40,41]) have highlighted greater male susceptibility to cortical atrophy and neuroinflammation under stress, which aligns with our findings of heightened pandemic-related brain ageing in males. These disparities underscore the potential interaction

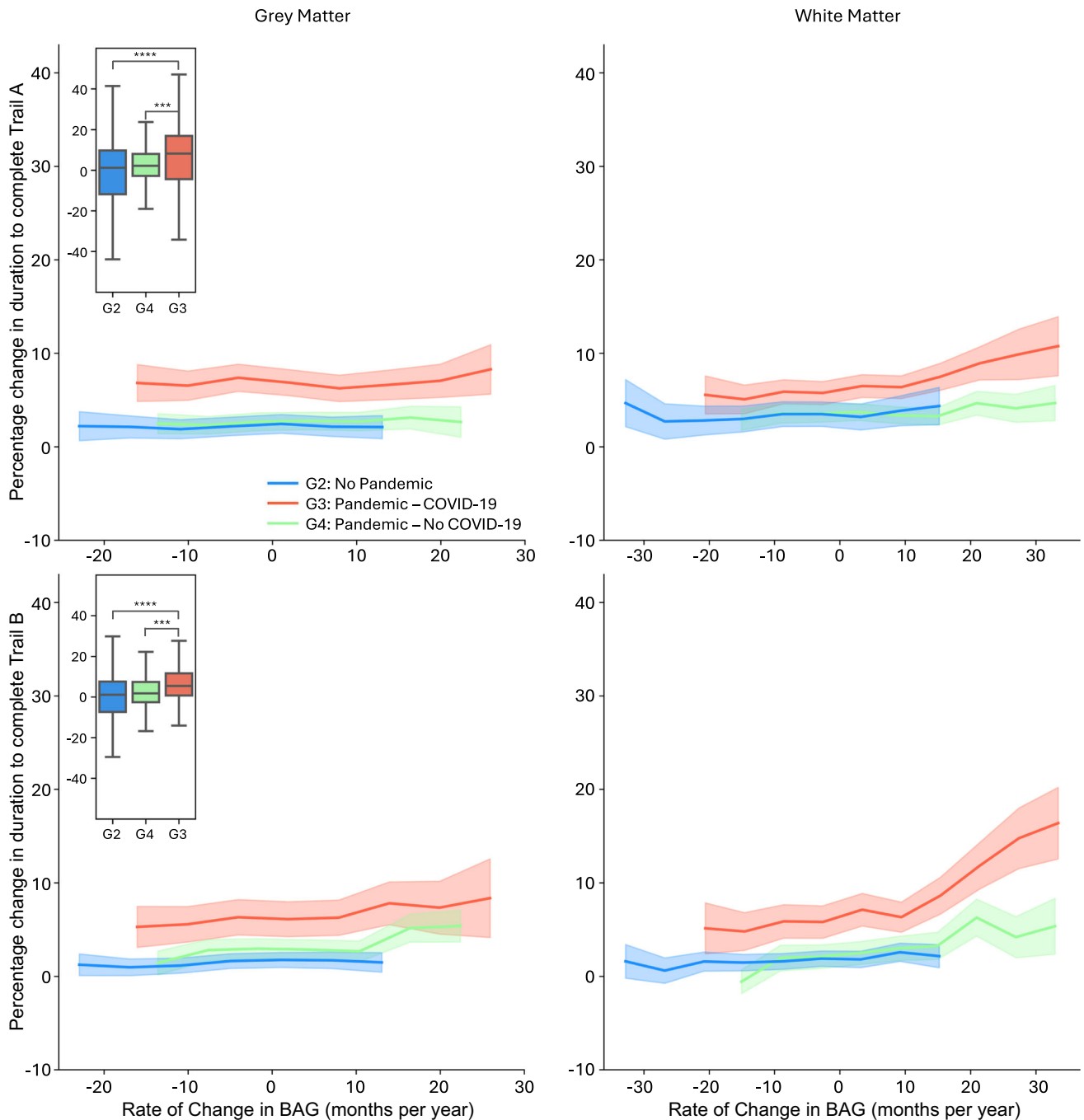

**Fig. 5 | Impact of COVID-19 on cognitive performance across rates of change in brain age gap.** The figure illustrates the percentage change in completion time for the Trail Making Test A (TMT-A, top row) and Trail Making Test B (TMT-B, bottom row) over two imaging time points across varying rates of change in brain age gap (BAG). Results are shown for the Pandemic–COVID-19 (G3, $N = 134$; red), Pandemic–No COVID-19 (G4, $N = 298$; green), and No Pandemic (G2, $N = 564$; blue) groups, using both grey matter (GM, left panels) and white matter (WM, right panels) models. A three-year sliding window was used to smooth the curves. Standard error is indicated using shaded areas: light blue (G2), light green (G4), and light red (G3). Boxplots (upper left of each row) display the raw distribution of percentage change in TMT performance, without a sliding window, for GM and WM models. Participants with COVID-19 (G3) showed greater decline in performance (i.e., longer completion times) compared to the Control group (G2), with FDR-corrected $p$-values of 1.0e-6 (TMT-A) and 9.1e-5 (TMT-B). Significant differences were also observed between COVID-infected (G3) and non-infected (G4) Pandemic participants (FDR-corrected $p$-values: 7.2e-4 (TMT-A) and 7.4e-4 (TMT-B)). Asterisks indicate statistical significance: *** denotes FDR-corrected $p \leq 0.001$; **** denotes FDR-corrected $p \leq 0.0001$. Group differences were assessed using two-sample $t$ tests.

between stress, sex-specific neural mechanisms, and accelerated ageing trajectories.

Our findings align well with reports of increased internalising symptoms, reduced cortical thickness, and accelerated brain ageing in adolescents during the pandemic[12]. However, in the middle-aged to older population we studied, advanced brain ageing is a direct indicator of poor brain health, without the complexities of adolescent brain maturation. A plausible explanation for the observed accelerated brain ageing is chronic stress, potentially linked to pandemic-related factors such as social isolation, economic insecurity, and health concerns, consistent with well-documented sequelae like neuroinflammation, structural and functional brain changes in preclinical

models[42–46]. Previous studies in humans confirm that social isolation and perceived loneliness contribute to structural and functional brain changes[47,48] that are expected to drive the observed accelerated brain ageing.

We further explored how pandemic-related stressors, including economic hardships, healthcare disruptions, and broader social inequalities, interact with pre-existing health disparities and age to influence brain health[49,50]. These stressors disproportionately affect vulnerable populations, worsening mental health challenges and amplifying socio-economic inequalities[51,52]. Our analysis revealed significant associations between deprivation and pandemic-related accelerated brain ageing, particularly in individuals with low employment, low education, and poor health scores. The observed interactions provide quantitative evidence for the differential effect of the COVID-19 on brain health across the population, with substantial widening of the brain age gap in socially and economically disadvantaged groups[53,54]. It remains unclear whether these brain ageing effects may be at least partially reversible, but the strong link to deprivation further emphasises the urgent need for policies addressing health and socio-economic inequalities, as the pandemic has exacerbated pre-existing disparities[37,55,56].

Previous studies have documented SARS-CoV-2's neural and vascular impacts, including inflammation and secondary systemic infection effects[22,57]. Our findings provide additional evidence of accelerated brain ageing in middle-aged to older participants with asymptomatic and mild-to-moderately affected COVID-19 infection without major comorbidities. This accelerated BAG effect was independent of infection status. Severity of COVID-19 infection, scan time since infection (Supplementary Fig. 1f), and the potential for long COVID are factors that could influence brain ageing results. However, as noted by Douaud et al. [2], the UK Biobank cohort predominantly comprises individuals with mild cases of COVID-19, which likely reflects a preselection of participants volunteering for re-imaging sessions as part of the UK Biobank recruitment approach. In our considered cohort, only 5 out of 134 participants (< 4%) required hospitalisation (Supplementary Fig. 1h), while the remaining participants experienced mild disease. Importantly, all participants tested negative within 2–3 weeks post-infection.

We showed a complex and partially differential effect of old age. While the BAG model was by design independent of chronological age, BAG change was higher in older age in all groups, including Controls, suggesting that age-related mechanisms contribute to the observed accelerated brain ageing[58]. This effect was strongest for the COVID-19-infected participants, which may offer an explanation for the observed differential effect on cognition. Cognitive decline is well-documented in ageing[35,59], and we confirm faster cognitive decline in older appearing brains in all groups. However, we report a distinctly more pronounced age effect in COVID-infected participants (with an apparent threshold in WM), suggesting a complex model of cognitive decline due to more pronounced accelerated brain ageing from infection-related factors in older people. This supports the concept of brain resilience loss leading to faster cognitive decline, consistent with existing neurodegeneration and dementia research[2,23,60,61] and recent epigenetic models[62,63].

It is conceivable that additional factors may have contributed to accelerated brain ageing during the pandemic[2,64] in both infected and non-infected subgroups such as reduced physical activity, poorer diets, and increased alcohol consumption, all negatively impacting brain health[65–67]. Our study focused on cumulative, easier to interpret brain ageing effects, limiting the ability to dissect region and modality-specific features that may disentangle diverse pathomechanisms. Nevertheless, differences in our findings derived from GM and WM models highlight potential implications for understanding neurodegeneration and other brain health issues[2,68]. Further research should explore specific GM and WM features driving acceleration of brain ageing that may allow to disentangle tissue, regional, and imaging marker-specific features that can be linked to neuro-glial-vascular mechanisms of brain ageing.

Our study has notable strengths and limitations. Employing BAG models provided an interpretable, brain-wide health marker that was sensitive to disentangle contributory biopsychosocial factors, leveraging the power of a longitudinal imaging-rich population study before and during the pandemic. We extended evidence on brain changes due to COVID-19 and socio-economic deprivation[2,37]. The subgroup comparisons highlight that the main brain 'cost' of the pandemic was not solely due to infection itself, though causal inference cannot be claimed[69]. More research is needed to clarify causal relationships between deprivation factors and accelerated brain ageing, considering complex interactions. The study is further limited by access to only two time points, prohibiting assessment of potential reversibility. Longer follow-ups after the pandemic are needed to investigate persistent brain ageing effects and their long-term consequences beyond acute cognitive impacts in the infected subgroup. Furthermore, a limitation of our study is the difference in interscan intervals (ISI) between the Pandemic and No Pandemic groups (with a wider spread in the Pandemic group compared with Controls), which can potentially influence effect estimation[33]. To minimise biases introduced by differences in ISI, we prioritised participants with longer follow-ups, as shorter intervals can introduce noise and increase susceptibility to outliers[25]. Furthermore, to account for potential biases related to differing inter-scan intervals, we applied multiple complementary statistical approaches tailored to different analyses. Across these approaches, including adjustments for age and normalisation by inter-scan intervals, we found that the results were stable and convergent, reinforcing the robustness of our findings. While ISI differences remain an inherent feature of the dataset, their impact on the reported effects appears minimal. Our study design deliberately excluded individuals with major mental health conditions, reducing the likelihood that pre-existing depression or anxiety influenced our findings. In addition, most available mental health data in the UKBB dataset were collected years before the pandemic, limiting their relevance to pandemic-specific effects. While we matched groups for household size (Supplementary Table 2)—an indirect measure of social contact and isolation—the considered deprivation indices did not capture state-dependent psychological stressors. Future research with longitudinal mental health data is needed to better understand the interplay between deprivation, stress, and brain ageing.

In conclusion, the COVID-19 pandemic profoundly impacted brain health, shown as accelerated brain ageing, influenced by bio-psycho-social factors, especially social and health deprivation. Notably, the main effects were independent of infection status, except for interactions between COVID-19 infection, brain ageing, old age, and cognitive decline. Our findings highlight the need to address health and socio-economic inequalities in addition to lifestyle factors to mitigate accelerated brain ageing. Continued research and targeted policies are crucial to improve brain health outcomes in future public health crises.

## Methods

### Study design and neuroimaging data

We drew participants from the UK Biobank (UKBB) imaging study, which provides multi-modal brain imaging data[19] from over 42,677 participants (released in April 2023), aged 45 and older. The UKBB has approval from the North West Multi-Centre Research Ethics Committee (MREC) to obtain and disseminate data and samples from the participants (http://www.ukbiobank.ac.uk/ethics/), and these ethical regulations cover the work in this study. Written informed consent was obtained from all participants. UKBB data were collected at four sites using identical protocols, ensuring consistency in imaging. Before the COVID-19 pandemic, approximately 3000 participants underwent a second imaging scan as part of a longitudinal study. Beginning in

February 2021, an additional 2000 participants were re-scanned to investigate the impact of SARS-CoV-2, bringing the total number of repeat scans close to 5000. Participants selected for re-imaging met specific criteria, including no incidental findings in their initial scans, residence within a defined catchment area, and high-quality imaging data at baseline.

In our study, to minimise confounding factors in brain age predictions, we excluded participants with chronic disorders[18,20] such as dementia, diabetes, heart and kidney disease, and depression (see full list in Supplementary Table 1), before both their first and second scans. This ensured a focus on healthy individuals and reduced potential biases associated with disease-related brain changes. The same exclusion criteria were applied consistently across both the training and unseen datasets. In addition, participants with low-quality anatomical MRI data[21] or unreliable brain IDPs were removed. Technical outliers—defined as IDP values exceeding five standard deviations from the cohort mean—and participants with substantial missing or unreliable IDPs in any session were also excluded.

For training the brain age prediction model[16], we included only participants with one imaging session collected before March 2020, ensuring they were free from chronic disorders or data quality issues, as detailed above (Fig. 1a, $N = 15,334$; 8407 female; age range: 45.1–82.4 years; mean ± SD: 62.6 ± 7.6 years). The trained model was applied to unseen groups of participants who underwent two imaging sessions ($N = 996$; 552 female; age range: 47.1–79.5 years; mean ± SD: 58.8 ± 6.2 years). These individuals were categorised into two main groups: The Pandemic group (G1), which included participants scanned both before and after the pandemic onset ($N = 432$; 255 female), and the No Pandemic group (G2, Control), consisting of individuals scanned twice before the pandemic onset ($N = 564$; 297 female). Within the Pandemic group, participants were further categorised into the Pandemic–COVID-19 group (G3), comprising individuals who contracted COVID-19 ($N = 134$; 78 female), and the Pandemic–No COVID-19 group (G4), consisting of those who did not contract the virus ($N = 298$; 177 female) (Fig. 1a). COVID-19 cases (G3) were identified using diagnostic tests, primary care records, hospital records, or antibody tests. To minimise potential confounding effects, all groups (G1–G4) were matched based on sex, age, BMI, alcohol intake, smoking, blood pressure, education, deprivation index, and general health metrics (Supplementary Table 2 and Supplementary Fig. 1).

The inter-scan intervals differed between groups, with a wider spread in the Pandemic group compared to Controls, due to the timings of the data acquired during the pandemic and lockdown interruptions (Supplementary Fig. 1c). The Control group had a narrow distribution of inter-scan interval around ~2.25 years, while for the Pandemic group shorter delays as low as 1 year could be found (Supplementary Fig. 1c). Motivated by ref. 25, which highlights the importance of longer follow-up durations in improving the reliability of detecting brain changes in longitudinal data, we kept participants with longer follow-ups and excluded the lowest 10% (i.e., < 2.0 years) of short-term follow-ups (Supplementary Fig. 1d). Changing this lower threshold from 10% to 20% did not change any patterns. Also, matching groups for mean instead of thresholding out low inter-scan intervals did not change any patterns in the results, but increased subjects with excessive BAG estimates – results available on preprint first version[70].

Participants' sex was determined based on self-report collected at enrolment and confirmed using genetic sex data provided by the UK Biobank. This ensured internal consistency across all participants. Sex was explicitly considered in the study design, with separate brain age models trained for males and females and sex-stratified analyses conducted where appropriate.

## Brain age modelling
We trained a multivariate regression model to estimate brain age by regressing imaging-derived phenotypes (IDPs) against participants'

ages. This resulted into an individual's brain age $Y_B$ and a brain age gap (BAG), defined as $\delta = Y_B - Y$, where $Y$ is the chronological age. A positive $\delta$ ($\delta > 0$) indicates an older-appearing brain, while a negative $\delta$ ($\delta < 0$) indicates a younger-appearing brain. Age was modelled as a function of $M$ imaging-derived phenotypes, $Y_B = f(\mathbf{X})$, with $\mathbf{X}$ being a matrix of dimensions $N \times M$, where $N$ is the number of participants. We used a general linear regression method introduced by Smith et al. [16] to ensure an unbiased $\delta$ orthogonal to chronological age.

Following established methodologies[18], separate models were trained for males and females, and for grey matter (GM) and white matter (WM), using IDPs derived from a healthy cohort free from chronic medical conditions (Fig. 1b). To mitigate dimensionality challenges inherent to large-scale imaging features, we applied singular value decomposition (SVD), retaining the top 50 components, consistent with the findings of ref. 16. This choice of 50 components balances model interpretability with predictive performance by capturing the majority of variance within the data, while minimising overfitting. The variance explained by these retained components was stable across cross-validation folds, as detailed in the Supplementary Materials, with minimal variation observed across models (e.g., mean variance explained for female GM: 57.8% ± 0.014, male GM: 57.9% ± 0.028, female WM: 79.0% ± 0.019, male WM: 78.7% ± 0.010). Reproducibility of patterns for different numbers of SVD components (from 30 to 100) was confirmed (Supplementary Fig. 8).

A 20-fold cross-validation process was used to train the model. In each iteration, a linear regression model was trained on 19 folds, and the fitted coefficients were applied to the held-out fold to predict brain age. During prediction, we de-confounded test set measures using the regressor's weights from identified confounding variables in the training set, following the approach used by Miller et al. [19]. Notably, age-dependent confounds were not removed from the IDPs. To ensure robustness, this cross-validation process was repeated 100 times with random assignment to folds in each trial, affirming the reliability of our brain age estimation model.

Post-training, the age prediction models were applied to unseen data from G1 and G2 groups, for both females and males (Fig. 1a). Predictions were performed independently for initial ($t_1$) and repeat scans ($t_2$) of participants, allowing estimation of brain age gaps $\delta_{t1}$ and $\delta_{t2}$, respectively. The rate of change in BAG ($R_{BAG}$) was then calculated as $(\delta_{t2} - \delta_{t1})/\Delta T$, where $\Delta T$ is the inter-scan intervals[18,31].

Unless otherwise stated, BAG values were aggregated across male and female participants within each group and tissue type for all group-level analyses.

## Feature selection for brain age modelling
To build predictive models for brain age estimation, we selected IDPs that focused on GM and WM regions (Fig. 1b). For the GM model, we used structural IDPs extracted from $T_1$-weighted MRI scans[19,21] which included measures such as the volume of subcortical structures, cortical/cerebellar regions, cortical surface area, cortical thickness, and GM/WM intensity contrast. This resulted in a total of $M = 1422$ IDPs. For the WM model, we used all WM-related IDPs derived from $T_1$-weighted data, a single IDP derived from $T_2$-weighted (total volume of WM hyperintensities), and IDPs derived from diffusion MRI scans, which reflected tissue complexity and integrity using diffusion tensor imaging (DTI) and neurite orientation dispersion and density imaging (NODDI) metrics. These metrics included fractional anisotropy (FA), mean diffusivity (MD), and eigenvalue maps, among others, resulting in 443 IDPs. CSF-related IDPs were excluded. Brain IDPs were obtained directly from the UK Biobank imaging data release and were used without any additional post-processing. These measures were generated using the UK Biobank's standardised image-processing pipeline, applied consistently across participants and imaging sites. Full details of the image acquisition, processing, and quality control procedures are described in Miller et al. [19] and Alfaro-Almagro et al. [21]. A complete

list of the selected IDPs included in our analysis is provided in Supplementary Tables 3 and 4.

All IDPs underwent de-confounding procedures to minimise the influence of potential confounders. This involved adjusting for 46 variables, including head size, sex, head motion, scanner table position, imaging centre, and scan-date-related drifts, while explicitly excluding age-related confounds to ensure unbiased brain-age predictions[19,71,72] (see Supplementary Table 5 for the full list of variables).

### Model performance evaluation

To assess model performance, we report the Pearson correlation coefficient (r) between chronological age and predicted brain age, as well as the mean absolute error (MAE) between these values. These metrics were averaged across all cross-validation folds. Specifically, we computed them over 100 repetitions of 20-fold cross-validation to ensure robustness.

### Modelling age effects in longitudinal brain ageing

To account for potential biases introduced by varying inter-scan intervals[33], $R_{BAG}$ was regressed against the average chronological age between the two scans rather than age at one of the time points. This approach has been used in previous longitudinal neuroimaging studies (e.g.,[18,31]) to provide a more balanced estimate of age-related effects on brain ageing.

### Interaction effects against socio-demographic factors

After calculating IDP-based brain age gaps and rates, we investigated interactions between brain ageing and socio-demographic factors using permutation-based inference with FSL PALM[73].

We conducted a series of 2-way analyses (permutation-based ANOVA) to examine the rate of change in BAG between two time points. Factor 1 was the pandemic presence; Factor 2 included socio-demographic variables: sex, regional employment, health, education, housing, and income scores. The latter five factors are indicators of deprivation (detailed descriptions of these indices can be found in the Supplementary Materials). Participants were categorised into 'high' and 'low' levels for each socio-demographic factor using the following thresholds: those scoring above the 70th percentile were classified as 'high', while those scoring below the 30th percentile were classified as 'low'. These percentiles were derived both from the entire UK Biobank population and separately based on the countries in which participants resided.

For each model tested, we assessed whether significant main effects existed—specifically, whether factor 1 (pandemic presence) or factor 2 (socio-demographic variable) had a discernible impact on brain ageing. In addition, we explored interaction effects to determine if the combined influence of both factors produced a different impact on brain aging compared to their individual effects.

### Cognitive scores

We selected the top 10 cognitive tests from the UKBB that have been associated with dementia risk[2] (see Supplementary Table 6). To compare participants' cognitive abilities across different groups, we calculated the percentage change in their cognitive scores between the two scans[2,74]. This was done using the formula: $Percentage\ change = (Score_{t2} - Score_{t1}) \times 100 / Score_{t1}$, where $Score_{t2}$ and $Score_{t1}$ represent the cognitive test results at the second and first time points, respectively.

To account for potential effects introduced by variations in inter-scan intervals (ISI)[33], we also tested an alternative normalisation method by dividing the change in score by the ISI. This was done using the formula: Rate of change $= (Score_{t2} - Score_{t1}) / \Delta T$, where $\Delta T$ is the inter-scan intervals.

### Reporting summary

Further information on research design is available in the Nature Portfolio Reporting Summary linked to this article.

## Data availability

All data used in this study were obtained from the UK Biobank (https://www.ukbiobank.ac.uk/), a large-scale biomedical database and research resource. The data are available to bona fide researchers through an application process and subject to UK Biobank's terms of access. Researchers can apply for access via the UK Biobank Access Management System (https://www.ukbiobank.ac.uk/enable-your-research/apply-for-access). This study was conducted under UK Biobank application number 43822 (PI: Stamatios Sotiropoulos).

## Code availability

Scripts for estimating brain-age using imaging-derived features and the pretrained models are available on GitHub https://github.com/SPMIC-UoN/BrainAge_COVID-19.

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

## Acknowledgements

Data were provided by the UK Biobank under Project ID 43822 (PI: S.N.S.). This study was supported by the NIHR Nottingham Biomedical Research Centre (grant to D.P.A., PI: Ian Hall) and the DEMISTIFI consortium, funded by the Medical Research Council under grant numbers MR/V005324/1 and MR/W014491/1 (R.G.J.). S.N.S. acknowledges support from the European Research Council (ERC Consolidator Grant 101000969). The computations described in this paper were performed using the University of Nottingham's Ada HPC service and the Precision Imaging Beacon Cluster, which provide High-Performance Computing support to the University's research community.

## Author contributions

A.M.: data curation, methodology, formal analysis, writing – original draft, writing – review & editing, M.C.: methodology, writing – review & editing, E.F.C.: writing – review & editing, X.C.: writing – review & editing, R.G.J.: funding acquisition, writing – review & editing, S.F.: funding acquisition, writing – review & editing, S.N.S.: methodology, resources, supervision, funding acquisition, conceptualisation, writing – original draft, writing – review & editing, D.P.A.: methodology, supervision, conceptualisation, funding acquisition, writing – original draft, writing – review & editing.

## Competing interests

R.G.J. reports research grants or contracts from AstraZeneca, Galecto, GlaxoSmithKline, Nordic Bioscience, Redx and Pliant, with all payments made to his institution. He has served as a consultant for AbbVie, AdAlta, Apollo Therapeutics, Arda Therapeutics, AstraZeneca, Brainomix, Bristol Myers Squibb, Chiesi, Cohbar, Galecto, GlaxoSmithKline, Mediar Therapeutics, Redx, Syndax and Pliant. He has received honoraria for lectures, presentations, speaker bureau participation, manuscript writing, or educational events from Boehringer Ingelheim, Chiesi, Roche and AstraZeneca. He has received payment for expert testimony from Pinsent Masons LLP. He has served on data safety monitoring boards or advisory boards for Boehringer Ingelheim, Galapagos and Vicore. He holds an unpaid advisory board role at NuMedii and serves as President of Action for Pulmonary Fibrosis. He is also Chair of the Editorial Board of BMJ Open Respiratory Research. All other authors declare no competing interests.
