## [Transparent Peer Review file · Nature Communications]

Accelerated Brain Ageing During the COVID-19 Pandemic

Corresponding Author: Professor Stamatios (N) Sotiropoulos

Version 0:

Reviewer comments:

Reviewer #1

(Remarks to the Author)

In the reviewed paper, the authors investigated the pandemic's impact on brain aging using longitudinal neuroimaging data from the UK Biobank. The authors used state-of-the-art multimodal modeling for assessing white and grey matter brain age and applied the model to two longitudinally scanned groups of individuals a) a "Control" group with both scans taken before the pandemic and the "Pandemic" group with one scan taken before and the other (during and) after the pandemic. The main results showed significant 1) accelerated brain aging for the pandemic group, and 2) which were independent of covid infection. They also showed 3) a correlation between accelerated brain aging and several socioeconomic and healthy indices within the pandemic group; 4) that older people within the pandemic group "brain aged" faster and 5) that accelerated brain aging correlated with reduced cognitive performance only in COVID-infected individuals. They discuss the findings in relation to the pandemic impact on brain health, beyond direct infection effects.

This is a very ambitious and thorough study, and the questions are very relevant thus the authors need to be commended for the effort. However, I have some major concerns with the main results as well as some important methodological concerns that I outline next:

(Major concerns)

1. Regarding the main control vs pandemic analysis. The authors found an "11-month higher deviation of brain age gap/year". This is such an extraordinary finding that seems too big to be true. It is difficult to see whether there is anything wrong with the data or the analyses (which seem correct as far as I can see) but based on indirect evidence from the report, the effect size of this analysis seems too big to be plausible. Let me try to elaborate: 1) theoretically, on a normal population, on average, one would not expect major changes in BAG across time (should be centered around 0). This is what the authors find in the control group, but not in the pandemic group which deviates a lot. 2) The authors report a high test-retest correlation, meaning that subjects with high BAG at baseline have high BAG at follow-up. This suggests that almost all individuals suffer an increase in BAG in a similar fashion (a common effect for all individuals). 3) This suggests the effect size of pandemic vs. control is enormous and extremely implausible (it would be essentially bigger than any effect reported on brain age). This reasoning is based on indirect evidence since the authors do not report effect sizes (years is not a valid effect size) (please, do report them), but I believe the authors should look into whether there is any covariate that may affect the analysis. I can only provide educated guesses: 1) the post-pandemic IDPs are generated from MRI data with different scanning parameters. My understanding is that the UKB pipeline uses FLAIR images to help with the reconstruction. While T1w images are identical, FLAIR was acquired using slightly different parameters. 2) Some possible issues with dimensionality reduction (PCA) which are poorly explained in the manuscript. My bet would go onto the first guess. This would be a serious problem for this study as the covariate would perfectly correlate with the variable of interest. It may be an unfair criticism as it is partially based on a hunch, but I really believe the authors should double-check the effect size of this result and possible explanations that are not centered on the effect of interest. As such, I believe the conclusion that "COVID-19 pandemic profoundly impacted brain health" should be put on hold before this concern is defused. 2) Effects of age on BAG change. The authors regressed RBAG against chronological age (at first scan). This is a methodological step that seems to work fine in randomized controlled trials, but can lead to completely wrong inferences when using observational data. If the spread of ISI is high, as it is in the present study, the risks are higher <https://journals.sagepub.com/doi/10.1177/0265407517718387>.

3) When computing cognitive performance, the authors use $tp2 - tp1 / tp1$ using the scores. This also represents a suboptimal methodological step. It does not seem to account for variations in time and it biases the scores by dividing them for $tp1$. Further, though it is an issue difficult to tackle it does not account for test-retest effects (this effect may be fixed, but

the decay is time-dependent which is problematic when having a significant spread in time for the interscanning interval).

(minor limitations)

- 4) please provide effect sizes (years is not a standardized effect size measure).
- 5) I understand why the authors remove individuals with longer follow-ups but in doing so they also remove the best quality data for capturing intraindividual change (Vidal-Pineiro, 2024). This is especially important when looking at differences within groups (e.g. cognition – BAG). The results seem to be driven by very extreme results (with RBAG > 10) which in turn are likely driven by individuals with short-term follow-ups that have unreliable estimates of change.
- 6) The introduction does not clarify the motivation for some analysis and methodological choices. Why do the authors separate WM and GM? What is the idea behind studying the relationship between BAG*age? Why do they choose the brain health and resilience factors that they choose? Please, provide a clear motivation in the introduction.
- 7) Multiple comparison corrections are needed.
- 8) Out of curiosity, there are >40K individuals with neuroimaging data at tp1. Why the training set was done with only 15K?
- 9) Please, explain better how dimensionality reduction was performed in both the test and training samples.

(Remarks on code availability)

Reviewer #2

(Remarks to the Author)

The study utilizes data from the UK Biobank to investigate cross-sectional and longitudinal differences in estimated brain age gap (BAG) associated with the COVID-19 pandemic. The study has a number of strengths including the large sample size, the multimodal brain age prediction model, the robust cross-validation procedure, the analysis of sociodemographic characteristics, and inclusion of cognitive performance. Overall this is an interesting, novel study that makes a significant contribution to our understanding of brain health after COVID-19, a topic which requires significantly more attention.

I do have several questions and comments:

I think that references 10 and 11 in the Introduction pg 3 are likely switched.

It would be useful for the authors to explain why they chose to create their own brain age prediction model rather than implementing an established one with known test-retest reliabilities.

I assume that the Pearson correlation coefficients reported for model performances are the mean coefficient across CV folds? This should be specified.

The model MAE's are lower than certain previously published brain age models. It would be useful to provide a more specific comparison. However, it should also be noted that the model built for this study has lower generalizability/usability than prior models given the complexity of IDPs required and the utilization of SVD with no details regarding the proportion of eigenvectors retained. In fact, I think these details should be provided in the supplement including the number of features retained and the total variance explained. The code provided by the authors suggests that the user select 10-25% of the eigenvectors. Additionally, the Results indicated that the authors retained "the top 50" eigenvectors that explained "the most" variance but the exact proportions for GM, WM x male female is not specified. How stable was the variance explained across different CVs of the training set?

The Results indicate "very high correlations between predicted brain ages of participants at the two time points" which is a very nice result that helps support reproducibility of the models. I assume these were Pearson correlations but intraclass correlations are probably better.

There are no results or discussion related to potential effects of factors such as COVID severity or time since infection on brain age (or cognition) in the infected subsample. It is also unclear whether any of the participants in the pandemic sample had PASC/long COVID which may influence the results significantly.

What is the explanation for the increased male vulnerability and is this potentially related to the higher prediction model MAE in males compared to females? I realize the difference in MAE was slight but any potential influence on these findings should be considered.

(Remarks on code availability)

The code is very well documented and I was able to install and run it. However, it should be noted that it does not provide their specific model for predicting brain age, i.e., it does not utilize the same IDPs (features) that the authors used to build their model but rather implements a generic model for predicting brain age from user specified IDPs. This is not a criticism but only a potential source of confusion for readers unless it was expected that the authors provide their actual model. If that is the case, then they did not provide it.

Reviewer #3

(Remarks to the Author)

This study investigated the influence of COVID-19 pandemic on brain ageing using longitudinal brain imaging data from the UK Biobank. This is an interesting and well-executed study. The manuscript is well-written. My concerns are relatively minor.

i) This study focused on healthy participants to avoid the impact of chronic illness on brain ageing. This is a valid and reasonable choice; however, it is unclear whether the 'absence of illness' criterion was only applied on baseline scans or if follow-up medical records were also obtained to exclude individuals who had developed some illness later (during or post pandemic) before the second scan. It is at odds that no one developed any other illness conditions even a subset group of individuals had covid-19 infection. The observed accelerated brain ageing in the pandemic group may be in part due to some disease effect other than pandemic, which needs to be disentangled.

ii) It is interesting that accelerated brain ageing is associated with the pandemic regardless of infection. It makes sense to explore social, economic and psychological factors to explain this observation. However, the insight provided by the deprivation indices alone is limited because the level of deprivation is very likely established before the pandemic, it does not provide a state-dependent estimate of stress level in people who have encountered the pandemic. It would be helpful to examine the association with psychological/distress measures, such as anxiety and depression symptoms and social isolation indices (from available questionnaires) and potentially the interaction with deprivation indices.

iii) The high correlation between predicted brain ages of participants at the two time points in figure 1e are likely driven by age and cannot indicate high scan-rescan model reproducibility.

iv) What is the cut-off for the high vs low deprivation score? Are these deprivation scores analysed from the baseline or follow-up?

(Remarks on code availability)

Version 1:

Reviewer comments:

Reviewer #1

(Remarks to the Author)

The authors have made a thorough job addressing my concerns and need to be commended.

I still believe this effects sizes are surprisingly big (way beyond what I would expect [but so are the effects of knee osteoarthritis pain on the brain]; so please: double-check/confirm no change in sequence, scanner upgrade, etc. happened between 2019-2021 in the UKB pipeline).

Given the authors double-check this, I am happy to endorse this manuscript for publication.

(Remarks on code availability)

Reviewer #2

(Remarks to the Author)

The authors have thoroughly addressed my prior concerns. I recommend the paper for publication.

(Remarks on code availability)

Reviewer #3

(Remarks to the Author)

The authors have properly addressed my concerns and I don't have further comments.

(Remarks on code availability)

Response to Reviewers' Comments

We would like to thank the reviewers for the helpful and considered reviews, which we believe have assisted us to substantially improve the manuscript. We have implemented most of the suggestions and we respond to each point in detail below.

The key changes in the manuscript main text have been highlighted in blue, as well as changed to figures and tables to reflect Reviewers' comments.

Reviewer #1

In the reviewed paper, the authors investigated the pandemic's impact on brain aging using longitudinal neuroimaging data from the UK Biobank. The authors used state-of-the-art multimodal modeling for assessing white and grey matter brain age and applied the model to two longitudinally scanned groups of individuals a) a "Control" group with both scans taken before the pandemic and the "Pandemic" group with one scan taken before and the other (during and) after the pandemic. The main results showed significant 1) accelerated brain aging for the pandemic group, and 2) which were independent of covid infection. They also showed 3) a correlation between accelerated brain aging and several socioeconomic and healthy indices within the pandemic group; 4) that older people within the pandemic group "brain aged" faster and 5) that accelerated brain aging correlated with reduced cognitive performance only in COVID-infected individuals. They discuss the findings in relation to the pandemic impact on brain health, beyond direct infection effects.

This is a very ambitious and thorough study, and the questions are very relevant thus the authors need to be commended for the effort. However, I have some major concerns with the main results as well as some important methodological concerns that I outline next.

We would like to thank the reviewer for their detailed and helpful review. We answer in detail each of their comments below.

Major concerns

1) Regarding the main control vs pandemic analysis. The authors found an "11-month higher deviation of brain age gap/year". This is such an extraordinary finding that seems too big to be true. It is difficult to see whether there is anything wrong with the data or the analyses (which seem correct as far as I can see) but based on indirect evidence from the report, the effect size of this analysis seems too big to be plausible. Let me try to elaborate: 1) theoretically, on a normal population, on average, one would not expect major changes in BAG across time (should be centered around 0). This is what the authors find in the control group, but not in the pandemic group which deviates a lot. 2) The authors report a high test-retest correlation, meaning that subjects with high BAG at baseline have high BAG at follow-up. This suggests that almost all individuals suffer an increase in BAG in a similar fashion (a common effect for all individuals). 3) This suggests the effect size of pandemic vs. control is enormous and extremely unplausible (it would be essentially bigger than any effect reported on brain age). This reasoning is based on indirect evidence since the authors do not report effect sizes (years is not a valid effect size) (please, do report them), but I believe the authors should look into whether there is any covariate that may affect the analysis. I can only provide educated guesses: 1) the post-pandemic IDPs are generated from MRI data with different scanning parameters. My understanding is that the UKB pipeline uses FLAIR images to help with the reconstruction. While T1w images are identical, FLAIR was

acquired using slightly different parameters. 2) Some possible issues with dimensionality reduction (PCA) which are poorly explained in the manuscript. My bet would go onto the first guess. This would be a serious problem for this study as the covariate would perfectly correlate with the variable of interest. It may be an unfair criticism as it is partially based on a hunch, but I really believe the authors should double-check the effect size of this result and possible explanations that are not centered on the effect of interest. As such, I believe the conclusion that “COVID-19 pandemic profoundly impacted brain health” should be put on hold before this concern is defused.

This is an important point and we would like to thank the Reviewer for providing so many thoughtful and really useful suggestions. The short answer is that we believe the solution is in comment 5 (again excellent suggestion by the Reviewer), and it has to do with imbalance between the groups in the follow-up duration between the two scans (Pandemic groups had long-term follow-ups compared to Controls). Adjusting for this, we still reproduce the originally reported trends, but effect sizes are now smaller (Cohen’s d in the order of ~ 1.2 originally is brought down to the 0.5-0.7 range). We redid all results and figures in the paper based on this adjustment, please see reply to comment 5.

For completeness, however, we first describe the work we did to exclude any other possibilities, as suggested in this comment. First, we are now reporting effect sizes using Cohen’s d and these were above 1 in the original submission, i.e. indeed large. There are not many studies on doing brain age prediction with longitudinal data. From the more recent ones, Tian and colleagues (Tian et al., *Nature Medicine*, 2023) report effect sizes of 0.5 to 1.1 when comparing BAG between major chronic diseases and healthy subjects, Zhao and colleagues (Zhao et al, *Nature Mental Health*, 2024) report effect sizes of 0.44 when comparing subjects with knee osteoarthritis pain and healthy subjects. So reported effect sizes from other studies have been medium to large, we would therefore expect large effects. Maybe not as high as our submitted results suggested, but big effects cannot be ruled out.

We conducted additional analyses to rule out other potential sources of bias, as suggested by the Reviewer. In doing so, we could exclude possibilities:

i) Normalisation of Δ BAG with Follow-Up Duration: Firstly, we investigated whether the normalisation of Δ BAG (the BAG difference between the two time points) by follow-up duration is causing any shift in the reported trends, given that follow-up periods were not perfectly matched between groups. A direct comparison of raw BAG differences between the two time points (as shown in Fig. P1b below) revealed similar trends to those originally reported (Fig. P1a, i.e. Fig. 2 of original submission).

ii) Scanner Parameter Differences (FLAIR Acquisition): Secondly, as correctly noted by the reviewer, there were minor changes mostly in the FLAIR acquisition protocols during the early phases of the UK Biobank imaging study (see page 11 of the protocol document: https://biobank.ctsu.ox.ac.uk/crystal/crystal/docs/brain_mri.pdf). Specifically, for the first ~ 500 participants (Phase 2, i.e. less than 1% of the total cohort), earlier versions of the scan protocols included slight modifications in sequence duration and intensity scaling of T2-FLAIR data. These protocols were shortened, by changing appropriate acquisition settings, in ways intended to not compromise contrast or image quality. In our study cohort, the vast majority of the considered subjects are from phases 3 and above. Specifically, the training cohort included 194 out of 15,334 (i.e. 1.26%) subjects from Phase 2, with no Phase 2 participants in the test cohort (i.e. all groups including Controls and Pandemic groups). The number in training cohort is small and we do not expect a significant change in the trends. Nevertheless, we retrained our Brain Age predictive models after excluding all Phase 2 participants from the training set and redid the BAG comparison between groups. The trends, comparing old (without excluding

subjects from Phase 2 – Fig. P1a) vs new (after excluding subjects from Phase 2 – Fig. P2) results, remain unchanged.

Fig. P1. Comparison of BAG differences between the two time points with and without normalisation for follow-up duration. (a) Figure 2 from the original submission, which presents BAG differences normalised for follow-up duration. (b) A direct comparison of raw BAG differences without normalisation, demonstrating similar trends to those observed in panel (a).

Fig. P2. Evaluation of the impact of early-phase scanner parameter differences on brain age gap estimates. This figure is showing the results of RBAG after removing all the participants who have been scanned in Phase 2 from all the groups.

iii) Impact of Dimensionality Reduction (PCA): Thirdly, we explored whether changing the number of PCA components retained changes significantly any trends. There is extensive discussion on the choice of 50 components in response to subsequent comments 9 and 13

below, nevertheless we tried 30 (lighter colours) and 100 (darker colours) components and compared against the 50 components. If we plot the rate of change in BAG between the two time points across different groups and for different number of components, the trends are mostly identical, both in grey and white matter, as shown in Figs. P3a-b. This figure is added to the Suppl. Materials as Suppl. Fig. S8.

Fig. P3. Effect of the number of retained principal components on the rate of change in brain age gap across different groups. Results are shown for (a) grey matter and (b) white matter models, comparing BAG rate of change when using 30, 50 (main model), and 100 principal components.

To summarise, we followed the Reviewer's suggestions to double-check and exclude the possibility that the observed trends and effect sizes are driven by nuisance parameters. We could not find any influence from the parameters mentioned in this comment, but please see reply to comment 5, as the mismatch in the follow-up duration between groups seems to be driving some of the reported sizes and considering this seems to provide a solution. We have revised the whole manuscript based on this.

2) Effects of age on BAG change. The authors regressed RBAG against chronological age (at first scan). This is methodological step that seems to work fine in randomized controlled trials, but can lead to completely wrong inferences when using observational data. If the spread of ISI is high, as it is in the present study, the risks are higher <https://journals.sagepub.com/doi/10.1177/0265407517718387>.

Thank you for pointing us to this useful reference Schilo & Grimm (2017). As follow-up intervals vary in our considered groups, regressing RBAG on age at first scan may introduce bias. To

address this, we have now used the average chronological age between the two scans ($AvgAge = (AgeT0 + AgeT1) / 2$) as the independent variable, rather than age at first scan. This provides a more balanced representation of an individual’s age trajectory over time and has been used in similar longitudinal studies (e.g., Tian et al., *Nature Medicine* 2023; Vidal-Pineiro et al., *eLife* 2021). This adjustment also helps mitigate potential statistical dependence between time points (Wainer, *The Centercept*, 2000).

As shown in Fig. P4a (i.e. Fig. 3a of the original submission), when using age at first scan, BAG acceleration was strongest in the “Pandemic – COVID-19” group, where each 1-year increase in baseline age corresponded to an acceleration of 13 days in GM and 16 days in WM. In comparison, for the “Pandemic – No COVID-19” group, the acceleration was 8 days in GM and 12 days in WM, while the Control group showed a smaller effect (5 days for GM and 7 days for WM). After adjusting for the average age between the two scans (Fig. P4b), the “Pandemic – COVID-19” group still showed the strongest effect, but BAG acceleration was now estimated at 10 days per year for both GM and WM, while the “Pandemic – No COVID-19” group exhibited 5 days for GM and 7 days for WM. Similarly, the effect in the Control group was reduced to 3 days for both GM and WM.

Since we have regenerated all figures based on reshaping our groups (see Comment 5), we include here Fig. P4c, which presents the same analysis as Fig. P4b but using the updated study cohort. Despite the adjustments, the overall trends remained unchanged, as in the original submission. Also, this consistent reduction in slope suggests that using average age may provide a more conservative estimate of brain ageing effects across different groups.

Please note that an alternative approach here would be to use the original model with $AgeT0$ and include follow-up duration (or interscan interval – ISI) as a covariate, i.e. $R_{BAG} = \beta_0 + \beta_1 \times Age(T0) + \beta_2 \times ISI$ (one would not use both ISI and the $AvgAge$, as that would be double-correcting). Results did preserve the same overall trends as in Fig. P4c, reinforcing the finding that slopes were higher in the Pandemic groups vs the No Pandemic.

Fig. P4. (a) Reproduction of the original submission’s Fig. 3, showing the relationship between chronological age at first scan and RBAG across different groups. (b) Updated analysis addressing the reviewer’s concern by replacing age at first scan with average chronological age $((AgeT0 + AgeT1) / 2)$ as the independent variable. (c) Same analysis as panel (b) but using the revised study cohort based on Comment 5. This updated cohort accounts for follow-up duration discrepancies, ensuring a more reliable assessment of brain ageing trajectories while preserving the observed trends.

In summary, we added a subsection titled “Modelling Age Effects in Longitudinal Brain Ageing” under the Methods to explicitly state that average chronological age was used instead of age at first scan, following recommendation from Tian et al. (*Nature Medicine* 2023) and Vidal-Pineiro et al. (*eLife* 2021). Additionally, the Results section now reports the updated regression models, effect sizes, and revised figures. In the Discussion, we address the mismatch of ISI between groups, referencing the Schilo & Grimm (2017) paper and the associated methodological challenges.

3) When computing cognitive performance, the authors use $tp2 - tp1 / tp1$ using the scores. This also represents a suboptimal methodological step. It does not seem to account for variations in time and it biases the scores by dividing them for $tp1$. Further, though it is an issue difficult to tackle it does not account for test-retest effects (this effect may be fixed, but the decay is time-dependent which is problematic when having a significant spread in time for the interscanning interval).

Thank you for the suggestion, we have followed the approach used in recent studies, including Douaud et al. (*Nature*, 2022) and Bloomberg et al. (*Nature Communications*, 2024), who both use exactly the same metric to assess longitudinal changes in cognitive function and we are now citing these papers in the relevant section in Methods.

However, we appreciate the reviewer’s comment that variations in inter-scan intervals could influence the observed effects. Firstly, we reanalysed the data using the suggested alternative that accounts for the ISI time interval between scans, i.e. instead of $(tp2-tp1)/tp1$ we plotted $(tp2-tp1)/ISI$. The trends remain consistent, as shown in Fig. P5b,d for both Trails A and B (the original submission figures are shown in Fig. P5a,c), even if some noisier behaviour was observed for Trail B. Nevertheless, the cognitive decline observed in individuals with COVID-19 infection persists regardless of the normalisation method used. Additionally, we continue to find no significant difference between the “Pandemic–No COVID-19” group and the “No Pandemic” group.

But none of these plots are in the new version of the manuscript, as we have now redone all analyses, based on Comment 5 (Fig. P9), which also preserves the trends. In the new version of the manuscript, we have kept both versions, i.e. both change with respect to first time point (main text – Fig. 5) and rate of change (in Supplementary Material – Fig. S6).

In summary:

- References have been added to the relevant Methods under “Cognitive Scores” section, to justify the choice of the normalisation method.
- The new Fig. 5 has been added to the main text and the alternative normalisation against follow-up interval has been added as Suppl. Fig. S6.
- A brief explanation in the Results section, stating that both methods produce similar trends

Fig. P5. Panels (a) and (c) reproduce Fig. 5 from the original submission, illustrating the percentage change in duration to complete Trails A and B, respectively. Panels (b) and (d) present the rate of change in duration to complete Trails A and B, accounting for variations in interscan intervals. In each row, the left-side plots correspond to the GM model, while the right-side plots correspond to the WM model. A three-year sliding window was used to generate the curves, with standard error values depicted in light colours. The inset boxplots in the top left of each row show the distribution of percentage (a, c) and rate of change (b, d) in TMT completion time without applying a sliding window. Statistical significance is indicated by **** for FDR-corrected p -values ≤ 0.0001 and *** for p -values ≤ 0.001 , calculated using two-sample t -tests between different groups.

Minor limitations

4) please provide effect sizes (years is not a standardized effect size measure).

Thank you, as described in point 1, we are reporting now Cohen's d throughout the manuscript and Figures.

5) I understand why the authors remove individuals with longer follow-ups but in doing so they also remove the best quality data for capturing intraindividual change (Vidal-Pineiro, 2024). This is especially important when looking at differences within groups (e.g. cognition – BAG). The results seem to be driven by very extreme results (with RBAG > 10) which in turn are likely driven by individuals with short-term follow-ups that have unreliable estimates of change.

We appreciate the reviewer's valuable feedback regarding the impact of excluding participants with longer follow-up periods on our findings. We acknowledge that by removing individuals with longer follow-ups, we may have unintentionally reduced the best quality data, as longer follow-ups capture meaningful intraindividual changes. Based on this important point, we carefully re-examined our dataset and performed additional analyses to better address the

reviewer’s concern. In fact, we found that adjusting for this discrepancy resolves to a large extent the concerns raised by the Reviewer in comment 1, and the very large effect sizes, as it looks that relatively short follow-ups in the Pandemic cohort were biasing estimates.

The distributions of the follow-up intervals are impossible to match between the Control and Pandemic groups in the UK Biobank. The Control group has a very narrow distribution (between 2 – 3 years) and very strong peak around 2.2 – 2.3 years between the two scans (Fig. P6a). For the Pandemic groups, due to the delays of the lockdowns, this distribution is very broad (between 1 – 7 years), yet there is significant density in the 2 – 3 year range (Fig. P6a). Our approach in the original submission was to threshold out the long follow-ups from the pandemic groups (> 3 years) and match the mean follow-up intervals across groups (Fig. P6b). However, as the Reviewer rightly pointed out, this is not necessarily optimal and a potentially better approach is to threshold out the shorter follow-up intervals (i.e., < 2 years), as less reliable that may be inflating effects. We therefore redid the analyses, adjusting for this and we tried in two different ways (Approach A and B), as shown in the Figure below (Figs. P6c-d). Note that in all histograms in Figs. P6b-d, the groups were also adjusted to be matched for age, sex, and all the other health markers reported in Suppl. Table S2.

Fig. P6. Histograms showing the distribution of inter-scan intervals across different methodological approaches. (a) The distribution of follow-up intervals in the available dataset. The Control group has a narrow distribution (2–3 years) with a strong peak around 2.2–2.3 years, while the Pandemic groups exhibit a broader range (1–7 years). (b) The approach used in the first submission, where follow-up intervals greater than 3 years were excluded to match the mean follow-up intervals across groups. (c, d) Two alternative approaches applied in the revised analysis: Approach A (excluding follow-up intervals < 2 years – lowest 10th percentile) and Approach B (excluding follow-up intervals < 2.1 years – lowest 20th percentile). All groups in panels b–d were further adjusted to be matched for age, sex, and relevant health markers (see Suppl. Table S2).

i) Approach A: Removing short follow-ups (remove lowest 10th percentile): Following the findings of Vidal-Piñeiro et al. (*bioRxiv* 2024), which highlight the importance of extended follow-up periods in improving the reliability of detecting brain changes, we revised our dataset. Specifically, we retained participants with longer follow-ups in the Pandemic group (who were excluded before) and excluded all the participants that were within the lowest 10th percentile of inter-scan intervals (i.e. participants with less than 2.0 years between scans – see Fig. P6c). The vast majority of the Controls have a tight distribution around 2 years, so in doing so, we excluded the extremely short follow-ups and matched this shortest follow-up interval. Subsequently, we matched the groups for age, sex, and other health markers. Please notice that because of this matching, the Control (i.e. No Pandemic G2 group) was also modified compared to the original submission (N = 564 from N = 932 originally).

The updated dataset that we use in the revised version now includes:

- G1: “Pandemic” group (N = 432; 255 female; age range: 47.1–79.5 years; mean \pm SD: 58.5 \pm 6.7 years),
- G2: “No Pandemic” group (N = 564; 297 female; age range: 48.1–72.3 years; mean \pm SD: 58.9 \pm 5.8 years),
- G3: “Pandemic – COVID-19” group (N = 134; 78 female; age range: 47.7–79.5 years; mean \pm SD: 58.7 \pm 7.0 years).
- G4: “Pandemic – No COVID-19” group (N = 298; 177 female; age range: 47.1–75.0 years; mean \pm SD: 58.4 \pm 6.5 years),

Figure P7 is a revised version of Fig. 2 of the original submission (also shown in Fig. P1a) and shows the revised rate of change in BAG (RBAG) across all groups and both brain tissue models for the new cohort. These results indicate that all differences and trends between groups reported before still remain, but there is a reduction in RBAG variability across all groups in both predictive models. However, the effect sizes (Cohen’s *d*) between groups, which initially ranged from 1.2 to 1.3 in the submitted version (for comparisons G2 vs G1, G2 vs G3, and G2 vs G4), have now decreased to a range of 0.5 to 0.7, suggesting a biologically more plausible medium effect size.

ii) Approach B: Removing even shorter follow-ups (remove lowest 20th percentile): To check whether the above results were robust to the choice of the low threshold for follow-up interval, we tried with setting a threshold higher to the 20th percentile (which was around 2.1 years – see Fig. P6d) and repeated the above analysis. Trends and effect size (Fig. P8) were the same as with the 10th percentile threshold, demonstrating convergence.

Fig. P7. Revised effect of COVID-19 and the pandemic on brain ageing based on Approach A. This figure presents the updated distribution of the rate of change in brain age gap (RBAG) across different brain tissue models and subject groups after applying a follow-up interval threshold at the 10th percentile (~2.0 years). While overall group differences and trends remain consistent with previous findings (Fig. P1a), RBAG variability has been reduced across all groups in both predictive models.

Fig. P8. Sensitivity analysis of the effect of COVID-19 and the pandemic on brain ageing using Approach B. This figure presents the updated distribution of the RBAG across different brain tissue models and subject groups using a higher follow-up interval threshold (20th percentile, ~2.1 years). The results remain consistent with those obtained using the 10th percentile threshold (Fig. P7), with similar trends and effect sizes across all groups, demonstrating the robustness of the findings to the choice of threshold.

In summary, and as approach B reduces further the total number of subjects that can be considered, we followed Approach A. We redid all analyses in the paper using this new definition of groups and we have revised all figures throughout. All main trends remain the same.

As part of these revisions, we have also revised Fig. 5 that the comment was originally aimed for (Fig. P9 and relevant Supplement). These results confirm that the general trends in group differences remain consistent. Specifically, the difference between the “Pandemic – No COVID-19” and “Pandemic – COVID-19” groups is still significant, further supporting our previous conclusions. With the revised analysis, we observed a reduction in the maximum range of RBAG values, resulting into narrower variability trends in the percentage changes in duration to complete Trails A and B.

To fully integrate these refinements, we have made the following changes to the manuscript:

1. Results Section:
 - The first paragraph has been modified to describe the revised approach to defining follow-up intervals.
 - All relevant subsections have been updated to reflect the impact of these changes on group comparisons and effect sizes.
 - All relevant figures in the main text have been updated accordingly.
2. Methods Section:
 - The “Study Design and Neuroimaging Data” subsection has been updated to describe the new follow-up interval thresholds and provide a justification based on recent literature (Vidal-Piñeiro et al., bioRxiv 2024).
3. Supplementary Materials:
 - Supplementary Figures and Tables have been revised to align with the updated dataset and analytical approach.
4. Discussion Section:
 - Revised to highlight the methodological improvements.

Fig. P9. Impact of including participants with longer follow-up on the relationship between cognitive performance and rates of change in brain age gap. This figure replicates the analysis presented in Fig. 5 of the main text but includes participants with longer follow-ups and removes those with the lowest 10% of short-term follow-ups (Approach A discussed above). The trends across the "Pandemic – COVID-19", "Pandemic – No COVID-19", and "No Pandemic" groups remain consistent with the main figure, with similar group differences and significance levels. However, the inclusion of participants with longer follow-ups reduces the maximum range of R_{BAG} values, providing a narrower variability in the percentage changes for TMT completion times.

Supplement to Fig. P9. Same as above, but instead of $(tp2-tp1)/tp1$, $(tp2-tp1)/ISI$ is plotted.

6) The introduction does not clarify the motivation for some analysis and methodological choices. (i) Why do the authors separate WM and GM? (ii) What is the idea behind studying the relationship between BAG*age? (iii) Why do they choose the brain health and resilience factors that they choose? Please, provide a clear motivation in the introduction.

Some of the methodological choices require a more detailed explanation. We feel expanding on certain methodological choices in the Introduction breaks the flow from the main messages and has the risk of focusing too much early on around technical points. We have therefore included relevant justifications in the methods and results sections (as early as first paragraph of results, just after introduction). Specifically, for each point:

i) Separation of WM and GM: Tissue-specific differential vulnerability to brain ageing and neurodegeneration is well established, with recent studies suggesting that COVID-19 may accelerate ageing processes differently in WM and GM due to different predominant biological mechanisms (Ellul et al., *The Lancet Neurology* 2020; Boldrini et al., *JAMA Psychiatry* 2021).

Neuroinflammation from COVID-19, for example, is expected to induce WM damage, accelerating WM ageing, disrupting connectivity, and increasing the risk of cognitive decline (Douaud et al., *Nature* 2022). In contrast, GM is more sensitive to hypoxia and direct neuronal invasion by neurotropic viruses such as SARS-CoV-2, with cortical GM particularly prone to age-related atrophy, which directly contributes to cognitive and functional impairments (Kremer et al., *Neurology* 2020). For these reasons, large-scale neuroimaging studies have modelled WM and GM separately when assessing neurodegenerative processes (Tian et al, *Nature Medicine* 2023), an approach we followed here.

While our primary motivation for modelling WM and GM separately stems from these neurobiological differences, we also acknowledge that broader environmental stressors introduced during the pandemic—such as lockdown-related deprivation and uncertainty—may have influenced brain ageing processes differently across tissue types. Although our study was not designed to isolate these effects, our results remain consistent with prior findings on tissue-specific vulnerabilities to ageing-related and disease-related changes.

In the first paragraph of the Results section, we added:

“As COVID-19 may affect differently WM and GM (Ellul et al., 2020; Boldrini et al., 2021; Douaud et al. 2022, Tian et al, 2023) and susceptibility to neurological diseases can vary across sexes (Nebel et al., 2018), separate models were trained based on GM and WM features, and for males and females (Tian et al., 2023)”

ii) Why study the relationship between BAG and chronological age: Age is the key risk factor for neurodegenerative processes, a range of age-related diseases and reduces protective and resilience factors in a non-linear fashion. Older adults typically experience greater declines in brain structure, including white matter integrity (and plasticity) and cortical thickness and show less adaptive brain function. They also are more susceptible to stress (Prenderville et al., *Trends in Neurosciences* 2015) and have impaired immune responses, suggesting heightened susceptibility to additional physical and mental stressors like COVID-19 or the pandemic (Fjell & Walhovd, *Nature Reviews Neuroscience* 2010; Raz & Rodrigue, *Neurosci Behav reviews* 2006). By studying BAG * age, we can identify age-specific vulnerabilities (Boyle et al., *Annals of Neurology* 2019).

To clarify this, we have now added the following sentence in Method section under “*Effects of age and sex on longitudinal brain ageing (rate of change in BAG)*” section which reads as follows: “*Understanding whether brain age acceleration varies across different age groups can reveal periods of increased vulnerability and potential dependencies on infection status and tissue specificity (Fjell & Walhovd, 2010; Raz & Rodrigue, 2006; Boyle et al., 2019).*”

iii) Choice of Resilience/Deprivation Indices: Our study incorporates key socioeconomic and demographic variables known to influence brain ageing trajectories, particularly in response to environmental adversity. Specifically, we selected deprivation indices related to education, health, employment, housing, and income based on their well-established roles as risk factors or moderators of brain ageing and age-related diseases (Livingston et al., *Lancet*, 2020; Farah, *Neuron* 2017; Hackman et al., *Nature Reviews Neuroscience* 2010). For example, lower education levels and poorer housing conditions have been linked to increased susceptibility to neurodegeneration (Farah, *Neuron*, 2017; Hackman et al., *Nature Reviews Neuroscience* 2010), while limited healthcare access and a greater disease burden correlate with accelerated brain ageing and increased susceptibility to adverse neurobiological outcomes (Zatorre et al., *Nature Neuroscience* 2012; Farah, *Neuron* 2017). Employment and income deprivation further contribute to chronic stress, nutritional deficiencies, and reduced healthcare access—factors that can amplify age-related changes in brain structure and function (Hackman et al., *Nature Reviews Neuroscience* 2010).

We therefore selected deprivation indices, including education, health, employment, housing, and income scores, to assess their role in shaping brain ageing trajectories and their potential impact on brain health in the context of the pandemic.

We have now clarified in the Results under “Increased brain age gap rate during pandemic in deprived areas” section:

“Besides age and sex, socio-demographic factors can influence brain health, cognitive reserve, and resilience to the detrimental effects of the pandemic (Livingston et al., 2020; Farah, 2017; Hackman et al., 2010).”

7) Multiple comparison corrections are needed.

We applied multiple comparison correction (MCC) using the False Discovery Rate (FDR) method across all relevant results. We apologise for not clearly documenting this in the original text. We have now clarified the application of FDR correction throughout the Results section and figure captions, ensuring that all reported p-values are FDR-corrected.

Additionally, we realised that MCC was initially omitted in the permutation-based tests for socio-demographic factors, and we have now addressed this as well. The results remain statistically significant, with only minor adjustments to the p-values that do not affect the main conclusions of the study.

8) Out of curiosity, there are >40K individuals with neuroimaging data at tp1. Why the training set was done with only 15K?

While over 40,000 individuals in the UK Biobank have neuroimaging data, our training set was deliberately limited to a set of 15,334 healthy participants to ensure that confounds from chronic illness are not affecting any of our results (see also Reply to R3-18). This allowed us

to train a “healthy normative” brain age model to compare against. A similar approach was followed in another recent study (Tian et al, *Nature Medicine* 2023).

Specifically, to create a robust brain age model that reflects healthy ageing patterns without confounding disease effects, we included only participants without chronic disorders (e.g., dementia, schizophrenia) before their imaging scans. We therefore selected healthy participants based on a comprehensive set of health criteria (see Supplementary Table S1) and excluded a large number of participants with a considerable range of chronic disorders similar to (Tian et al., *Nature Medicine* 2023, Massen et al., *QJM* 2023). This approach enabled us to develop a model that aims to predominantly capture healthy brain ageing, thereby ensuring biological meaningful deviation scores and validity in prediction when applied to our study cohort (who also had the same chronic illness exclusion criteria applied at both timepoints of all participants).

We also excluded subjects with incidental findings and particularly bad neuroimaging data quality (significant motion artifacts and extreme outliers in imaging-derived phenotypes (IDPs) ≥ 5 standard deviations from the mean). However, these were less than 1% of the total excluded subjects; subjects due to the former reason (chronic illnesses) comprised more than 99% of the total number of excluded participants.

We have updated the manuscript to emphasize these exclusion criteria in the first paragraph of Results and in the “*Study Design and Neuroimaging Data*” section of Methods.

9) Please, explain better how dimensionality reduction was performed in both the test and training samples.

As requested, we have provided further detail throughout, both in the main text and Supplementary Materials. In summary:

i) Dimensionality Reduction in Training Data: For each cross-validation fold, we applied Singular Value Decomposition (SVD) to the IDPs of the training data, breaking them down into orthogonal components ranked by their variance contribution. We chose to retain the top 50 components based on recommendations by Smith et al. (*NeuroImage* 2019), as an optimal choice for brain age modelling and brain age gap prediction using UK Biobank imaging-derived features. This strikes a balance between capturing the core variability in the data and maintaining model dimensionality, thereby reducing the risk of overfitting (see reply to comment 1 (Fig. P3), showing that change of this number from 30 to 100 components, does not change trends).

ii) Projecting the Test Data: To maintain a consistent feature space for both training and test data, we projected the test data onto the same 50-component space derived from the training data’s SVD. This approach maintains a uniform feature representation across datasets, preserving model independence and preventing data leakage. The above two steps have been added to the Suppl. Materials under the “*Dimensionality Reduction Using Singular Value Decomposition (SVD)*” section.

iii) Stability during cross-validation: Our entire dimensionality reduction process was embedded within a 20-fold cross-validation framework. For each fold, the SVD basis was recalculated solely from the training data, ensuring that the test data remained independent. This process consistently demonstrated stable variance explained by the retained 50 components (e.g. across all folds, variance explained was roughly 58% and 79% for GM and WM model, respectively, with minimal variation). These details have been added to the

Methods section under “Brain Age Modelling” which reads as follows: “To mitigate dimensionality challenges inherent to large-scale imaging features, we applied singular value decomposition (SVD), retaining the top 50 components, consistent with the findings of ¹⁶. This choice of 50 components balances model interpretability with predictive performance by capturing the majority of variance within the data, while minimising overfitting. The variance explained by these retained components was stable across cross-validation folds, as detailed in the Suppl. Materials, with minimal variation observed across models (e.g., mean variance explained for female GM: 57.8%±0.014, male GM: 57.9%±0.028, female WM: 79.0%±0.019, male WM: 78.7%±0.010). Reproducibility of patterns for different number of SVD components (from 30 to 100) was confirmed (Suppl. Fig. S8).”

Reviewer #2

The study utilizes data from the UK Biobank to investigate cross-sectional and longitudinal differences in estimated brain age gap (BAG) associated with the COVID-19 pandemic. The study has a number of strengths including the large sample size, the multimodal brain age prediction model, the robust cross-validation procedure, the analysis of sociodemographic characteristics, and inclusion of cognitive performance. Overall this is an interesting, novel study that makes a significant contribution to our understanding of brain health after COVID-19, a topic which requires significantly more attention.

I do have several questions and comments.

10) I think that references 10 and 11 in the Introduction pg 3 are likely switched.

Thank you for catching this! We have now corrected and reversed the order of these two references (2nd paragraph of the Introduction section).

11) It would be useful for the authors to explain why they chose to create their own brain age prediction model rather than implementing an established one with known test-retest reliabilities.

Even if we understand the motivation behind the reviewer's comment, we are not aware of a "defacto standard" brain age prediction model that is established and widely accepted for general use. By creating our own model, we minimised potential confounds associated with many nuisance factors. Specifically:

i) Lack of harmonisation in neuroimaging. Due to the lack of standardisation (harmonisation) in neuroimaging data and the high variability of imaging-derived features across scanners, UK Biobank imaging data (K. Miller et al., *Nature Neuroscience* 2016) were purposefully acquired on a few identical scanners with standardised protocols, resulting in a dataset that is highly harmonised by design. Hence, and given the number of participants of the UK Biobank, it would have been a missed opportunity not to use these data to train a model and avoid potential neuroimaging confounds (e.g., scanner variations, protocol differences) that may arise from integrating data from other studies. In addition, the rich set of features that can be extracted from the UK Biobank are not always supported by more conventional studies, hence a simpler brain age prediction model would have potentially led us to discard lots of data and modalities.

ii) Choosing a "Normative" Cohort for Training: Relevant to points 8 and 18 raised by Reviewers 1 and 3, respectively, we excluded a large number of participants that had chronic illnesses to avoid having relevant confounds in our results. By training using data from healthy participants, excluding ones with chronic disorders, we effectively created a normative healthy brain age model. Subsequently, we matched training and test groups (at both time points) for health status (i.e. excluded participants with chronic disorders from all groups), to ensure that results largely reflect the impact of pandemic.

iii) Availability of brain-age predictions through UK Biobank: Even if recent studies have used UK Biobank to train brain age prediction models (e.g., Smith et al. *NeuroImage* 2019, Smith et al. *eLife* 2020, Tian et al. *Nature Medicine* 2023, Richard et al. *Hum. Brain Mapp* 2018, and Cole *Neurobiol. Aging* 2020), UK Biobank limits how these are shared publicly. According to the material transfer agreement, results need to be returned to them, and they release them to approved projects. This induces a lag, and at this point, there are no released

brain age predictions from previous studies available for UK Biobank data in healthy participants (but very likely, these would be in the future). Having said that (Peng, *Medical Image Analysis*, 2021), utilised a deep learning approach and released a trained brain age prediction model (so provides these estimates indirectly). Crucially, however, this is not trained exclusively on healthy participants, making it unsuitable for our study (see point B above).

iv) Large Training Cohort: Our model was trained on tens of thousands of UK Biobank participants (other studies have used considerably fewer subjects (Zhao et al. *Nature Mental Health* 2024, Richard et al. *Hum. Brain Mapp* 2018, and Cole *Neurobiol. Aging* 2020), providing a substantial sample size to enhance model robustness and generalisability within this dataset. This large, homogeneous training set also enhances the model's relevance for longitudinal studies within the UK Biobank, supporting reliable predictions for future analyses of ageing trajectories.

v) Addressing Age-Related Bias: We adopted a methodological approach that removes bias of the brain age delta against age (Smith et al, *Neuroimage* 2019), an issue that can notably skew brain age predictions. This bias-correction step is essential to avoid under- or overestimating brain age effects in specific age groups, which is particularly important in studies investigating accelerated ageing patterns due to COVID-19.

We have added a paragraph in the Discussion that overviews these points: *“For the brain age prediction models, we relied on neuroimaging data exclusively from the UKBB (K. Miller et al., Nature Neuroscience 2016), hence minimising potential confounds from scanner variability and protocol differences inherent to datasets pooled across different studies. At the same time, we leveraged the rich set of imaging-derived features available in thousands of UKBB participants. By training using only healthy participants without chronic disorders, we effectively constructed normative brain age models with a substantial sample size. Importantly, we employed a bias-correction step to ensure that brain age delta was independent of chronological age, mitigating potential skewing effects that could influence ageing estimates (S. Smith et al. (NeuroImage 2018), De Lange and Cole (Neuroimage Clinical 2020), Zhao et al. (Nature Mental Health 2024)).”*

12) I assume that the Pearson correlation coefficients reported for model performances are the mean coefficient across CV folds? This should be specified.

Indeed, the Pearson correlation coefficients are the mean across all cross-validation folds. This was mentioned in the caption of Figure 1, *“Accuracy is evaluated using Pearson correlation coefficient (r) and mean absolute error (MAE), averaged over 100 repetitions of 20-fold cross-validation”*, but we appreciate it can be missed.

We have now added an explicit description in Methods, under *“Model Performance Evaluation”* section, emphasising that the reported metrics are averaged across all cross-validation folds.

13) The model MAE's are lower than certain previously published brain age models. It would be useful to provide a more specific comparison. However, it should also be noted that the model built for this study has lower generalizability/usability than prior models given the complexity of IDPs required and the utilization of SVD with no details regarding the proportion of eigenvectors retained. In fact, I think these details should be provided in the supplement including the number of features retained and the total variance explained. The code provided by the authors suggests that the user select 10-25% of the eigenvectors. Additionally, the Results indicated that the authors retained

“the top 50” eigenvectors that explained “the most” variance but the exact proportions for GM, WM x male female is not specified. How stable was the variance explained across different CVs of the training set?

Thank you for these comments, as there are a few points raised, we are addressing each of them separately:

i) Lower MAE in our brain age-prediction model compared to other studies

We respectfully disagree, our MAEs are in par with previously published studies that used the same unbiased estimation methodology. We have used a brain-age prediction approach that orthogonalizes the predicted brain age gap to chronological age, therefore ensuring unbiased brain-age estimation (Smith et al., *Neuroimage* 2019). Smith et al. highlighted that for biased age prediction models, MAEs ranged from 3.6 to 5.6 years, while MAEs for the unbiased models were in the 2.2 to 3.0 years range. De Lange and Cole (*Neuroimage Clinical*, 2020) also confirmed that accounting for this orthogonalization step results in lower MAEs. Our MAEs ranged from 2.9 to 3.08 years with stable predictive performance across cross-validation folds and similar values to Smith et al. (*NeuroImage* 2019), who for similar models and feature dimensionality in the UK Biobank reported MAE around 2.9 years.

A number of other studies have used the same unbiased approach and have also reported lower MAEs than the ones reported before for biased models. Peng et al. (*Medical Image Analysis*, 2021) have reported MAE of 2.8 – 3.3 years when using a general linear model approach (as we do), and Zhao et al. (*Nature Mental Health*, 2024) MAE of 2.37 years when using an elastic nets regression approach.

In summary, our reported MAEs are lower than MAEs reported by biased models and are in agreement and as expected by various recent papers that perform a similar unbiased brain age prediction. To address this part of the comment, a new paragraph is added to the Discussion section reads as follows:

“In our study, we achieved high accuracy and low MAE in brain age prediction, aligning with prior unbiased methodologies. For instance, Smith et al. (2019) reported MAEs of ~2.9 years using orthogonalized brain age gap prediction models, consistent with the MAEs observed in our study (2.9–3.08 years). This unbiased approach, as further supported by Peng et al. (2021) and Zhao et al. (2024), consistently achieves lower MAEs compared to biased models, as highlighted by de Lange and Cole (2020). These findings confirm the robustness of our model and demonstrate its comparability to state-of-the-art methods in the field.”

ii) IDP complexity

The IDPs used in our study are derived from relatively standard, widely accessible imaging modalities and are the same as used in numerous recent brain age prediction studies (e.g., Smith et al. *NeuroImage* 2019, Smith et al. *eLife* 2020, Richard et al. *Human Brain Mapp* 2018, Cole *Neurobiology of Aging* 2020, Tian et al. *Nature Medicine* 2023, Xiong et al. *Sensors* 2023, Bintsi et al. *MICCAI* 2023). They are extracted from relatively bog-standard data available in all clinically approved 3T scanners nowadays, including standard T1w and T2w FLAIR and standard resolution diffusion MRI. The IDPs can also be generated using publicly available code, which comes as a prepackaged docker container (i.e., in a way as easy to setup as, for instance, to extract cortical areal volumes in a T1w image).
<https://www.fmrib.ox.ac.uk/ukbiobank/fbp/>
https://git.fmrib.ox.ac.uk/falmagro/UK_biobank_pipeline_v1.

To address this part of the comment, a few sentences are added to Methods under the “*Feature Selection for Brain Age Modelling*”:

“Brain IDPs were obtained from the UK Biobank release (i.e. no extra processing was performed to extract those from the brain MRI data). The IDPs are generated and released as described in Alfaro-Almagro et al., NeuroImage 2018, utilising a multi-modal processing pipeline, publicly available and packaged as a Docker container.”

iii) SVD Utilisation and Dimensionality Reduction

We have added further details in response to similar comments 1 and 9 by Reviewer 1. For the models presented in our study, we employed SVD for dimensionality reduction (similar to Fong et al., *Nature Aging* 2024, Baecker et al., *Human Brain Mapping* 2021, Monti et al., *PLOS ONE* 2020, Xifra-Porxas et al., *NeuroImage* 2021, Sihag et al., *IEEE Journal of Selected Topics in Signal Processing* 2024), retaining the top 50 components as a balance between predictive performance and model interpretability, in line with findings from Smith et al. (*NeuroImage* 2019) on the UK Biobank data. The GM and WM models comprised 1,422 and 433 features, respectively, so with 50 SVD components, 4% and 11% of the total eigenvectors were kept respectively. The total variance explained by these features was ~58% for the GM features and ~79% for the WM features, for both males and females. These are also in par with previous studies (Franke et al. *NeuroImage* 2010). To ensure the generalisability of trends, we explored and confirmed replication of our results using 30 and 100 components as well, demonstrating robustness of identified trends to the choice of the number of components. These are now presented in Fig. P3 (Suppl. Fig. S8).

To address this part of the comment, a new paragraph is added to the Method section under the “*Brain Age Modelling*” reads as follows (A new section is also added to the Suppl. Materials):

“To mitigate dimensionality challenges inherent to large-scale imaging features, we applied singular value decomposition (SVD), retaining the top 50 components, consistent with the findings of Smith et al. (2019). This choice of 50 components balances model interpretability with predictive performance by capturing the majority of variance within the data, while minimising overfitting. The variance explained by these retained components was stable across cross-validation folds, as detailed in the Suppl. Materials (Suppl. Fig. SX), with minimal variation observed across models (e.g., mean variance explained for female GM: 57.8% ± 0.014, male GM: 57.9% ± 0.028, female WM: 79.0% ± 0.019, male WM: 78.7% ± 0.010). Reproducibility of patterns for different number of SVD components (from 30 to 100) was confirmed (Suppl. Fig. S8).”

D. Stability Across Cross-Validation Folds

Across 20-fold cross-validation, the variance explained by the first 50 components were stable in WM and GM models, both for female and male. Specifically, for Female GM across folds, variance explained = 57.8%±0.014; for Male GM, variance explained = 57.9%±0.028; for Female WM, variance explained = 79.0%±0.019; for Male WM, Mean variance explained = 78.7%±0.010. To address this part of comment, these values are now reported in the Suppl. Material under “*Dimensionality Reduction Using Singular Value Decomposition (SVD)*” section and in the Method section under “*Brain Age Modelling*” (discussed above).

14) The Results indicate “very high correlations between predicted brain ages of participants at the two time points” which is a very nice result that helps support reproducibility of the models. I assume these were Pearson correlations but intraclass correlations are probably better.

Thank you for your suggestion to focus on ICC. We have calculated the intraclass correlation coefficients (ICCs) to support the reproducibility of our brain age predictions across time points. In the Results section, under "*Performance of brain age prediction models*", we now report an ICC of 0.981 (95% CI: 0.977–0.985) for the "Pandemic" group and 0.983 (95% CI: 0.980–0.985) for the "No Pandemic" group, indicating stability of estimated brain ages over time. We also updated Fig. 1e to reflect these ICC values, providing a clearer representation of model reproducibility.

15) There are no results or discussion related to potential effects of factors such as COVID severity or time since infection on brain age (or cognition) in the infected subsample. It is also unclear whether any of the participants in the pandemic sample had PASC/long COVID which may influence the results significantly.

Unfortunately, there are only limited data available in the UK Biobank with respect to these factors. However, there is overall low variability in COVID severity in the longitudinal UK Biobank cohort we considered. As requested, we have added a relevant section in Discussion, summarising the points detailed below.

Specifically, within the considered COVID-19 cohort of 121 participants (i.e. participants with longitudinal scans, who tested positive for COVID-19), only five individuals were hospitalised (this information is added as a new figure as Suppl. Fig. S1f – Fig. P10c), indicating that the majority of the group experienced a milder clinical course of COVID-19. We re-analysed the data after excluding these hospitalised participants, and the results remained unchanged. Specifically, the distributions of the rate of change in brain age gap for the full cohort (i.e. including hospitalised subjects, Fig. P1a) and the cohort excluding hospitalised participants were highly similar, across both GM and WM models.

Regarding the potential impact of time since infection, Suppl. Fig. S1f (Fig. P11b) illustrates the distribution of time intervals between infection and the follow-up brain scan. The majority of participants were scanned within 2 to 6 months post-infection. Importantly, all participants tested negative for SARS-CoV-2 approximately two to three weeks following their initial diagnosis. In addition, anyone developing a new health related condition related or unrelated to COVID-19 was excluded from the analysis.

As for the possible influence of PASC/long COVID, we found no evidence or data of participants meeting the criteria for prolonged symptoms or requiring more intensive medical interventions. Specifically, none of the participants had records indicating admission to an intensive care unit, or the use of invasive or non-invasive ventilation, continuous positive airway pressure (CPAP), or unspecified oxygen therapy.

Taken together, these findings suggest that our cohort predominantly consists of individuals with mild COVID-19 who recovered without significant medical intervention. This pattern likely reflects the UK Biobank's voluntary re-imaging recruitment strategy, as previously noted by Douaud et al. (*Nature*, 2022).

The new paragraph in the Discussion reads as follows:

“Severity of COVID-19 infection, scan time since infection (Suppl. Fig. S1f), and the potential for long COVID are factors that could influence brain ageing results. However, as noted by Douaud et al. (2022), the UK Biobank cohort predominantly comprises individuals with mild cases of COVID-19 which likely reflects a preselection of participants volunteering for re-imaging sessions as part of the UK Biobank recruitment approach. In our considered cohort, only 5 out of 134 participants (<4%) required hospitalisation (Suppl. Fig. S1h), while the remaining participants experienced mild disease. Importantly, all participants tested negative within 2–3 weeks post-infection.”

Fig. P10. (a) Rate of change in brain ageing gap between the full Pandemic – COVID-19 cohort and after excluding hospitalised participants. (b) Distribution of the time interval (in months) from the first confirmed positive COVID-19 test to the repeat brain imaging scan, illustrating scan times post-infection. (c) Distribution of hospitalised versus non-hospitalised COVID-19 cases within the Pandemic – COVID-19 group.

16) What is the explanation for the increased male vulnerability and is this potentially related to the higher prediction model MAE in males compared to females? I realize the difference in MAE was slight but any potential influence on these findings should be considered.

Sex differences have been reported before in susceptibility to ageing. For instance, Gur and Gur, *Neuroscience and Biobehavioral Reviews* (2016) and Armstrong et al., *Neurobiology of Aging* (2019) demonstrated sex differences in brain structure and function with age, showing that males are more likely to exhibit accelerated brain volume loss and cognitive decline in ageing populations. Even if the mechanisms are likely different, the higher male vulnerability to brain pathology is compatible with higher male vulnerability to an increased brain age gap due to the COVID-19 pandemic. We have included in Discussion a relevant sub-section.

In addition, we conducted an exploratory analysis to further investigate whether the observed higher male susceptibility to pandemic was linked to the higher MAE in brain age prediction models for males compared to females. To do so, we identified two other male subgroups A and B with different MAEs and explored whether higher MAE is also linked to higher susceptibility (i.e. as observed in the male-female example) or whether that relationship no

longer holds. For group A, we considered males coming from a deprived background, and MAE was 2.85 years. For group B, we considered males coming from a non-deprived background, and MAE was higher at 3.12 years. However, group B (that had the higher MAE) was found to be less susceptible to the pandemic effects (i.e. opposite trend to the males-females case). Even if such an example does not rule out MAE differences having some effect in the observed trends, it does rule out the presence of a generic predictive relationship between high MAE and high susceptibility to the pandemic.

The new paragraph in the Discussion reads as follows:

“Male vulnerability to brain ageing was particularly pronounced during the pandemic, consistent with prior evidence of sex differences in neurobiology. Studies (e.g., Nebel et al. 2018; Gur and Gur 2016) have highlighted greater male susceptibility to cortical atrophy and neuroinflammation under stress, which aligns with our findings of heightened pandemic-related brain ageing in males. These disparities underscore the potential interaction between stress, sex-specific neural mechanisms, and accelerated ageing trajectories.”

Reviewer #2 (Remarks on code availability):

17) The code is very well documented and I was able to install and run it. However, it should be noted that it does not provide their specific model for predicting brain age, i.e., it does not utilize the same IDPs (features) that the authors used to build their model but rather implements a generic model for predicting brain age from user specified IDPs. This is not a criticism but only a potential source of confusion for readers unless it was expected that the authors provide their actual model. If that is the case, then they did not provide it.

We are sorry for the confusion. To clarify, we have now added our pretrained models to the GitHub repository. These models directly reflect the approach used in our study and are designed to different groups, including separate models for males and females, as well as models specifically trained on grey matter and white matter:

- 1- Trained_model_50PCs_wm_M,
- 2- Trained_model_50PCs_gm_M,
- 3- Trained_model_50PCs_wm_F,
- 4- Trained_model_50PCs_wm_F.

By making these models available, we aim to provide users with a more direct way to apply our approach without the need to define their own IDPs. The pretrained models can be accessed at the following link:

https://github.com/SPMIC-UoN/BrainAge_COVID-19/tree/main/analysis/trained_models

Additionally, we have updated the “Data Sharing” section to reflect the availability of the pretrained models in the GitHub repository.

Reviewer #3

This study investigated the influence of COVID-19 pandemic on brain ageing using longitudinal brain imaging data from the UK Biobank. This is an interesting and well-executed study. The manuscript is well-written. My concerns are relatively minor.

Thank you for the positive feedback, we have addressed your comments below.

18) This study focused on healthy participants to avoid the impact of chronic illness on brain ageing. This is a valid and reasonable choice; however, it is unclear whether the 'absence of illness' criterion was only applied on baseline scans or if follow-up medical records were also obtained to exclude individuals who had developed some illness later (during or post pandemic) before the second scan. It is at odds that no one developed any other illness conditions even a subset group of individuals had covid-19 infection. The observed accelerated brain ageing in the pandemic group may be in part due to some disease effect other than pandemic, which needs to be disentangled.

We confirm that we excluded all participants with chronic disorders prior to both their first and second scans in both the training and test datasets, ensuring that only healthy individuals were included throughout the study. This reduced the number of subjects considered in our study design (see also reply to R1-8), as we effectively considered “super-healthy” groups for all group comparisons, excluding a range of chronic disorders as in Tian et al. *Nature Medicine* 2023 and Massen et al. *QJM* 2023. This approach minimised potential confounding effects of chronic illnesses on brain age estimation. We had access to comprehensive medical records before and after each scan, allowing us to monitor and confirm participants' health status at both time points.

To clarify this process, we have revised the manuscript text in both the Results, page 5 and Methods, page 25.

19) It is interesting that accelerated brain ageing is associated with the pandemic regardless of infection. It makes sense to explore social, economic and psychological factors to explain this observation. However, the insight provided by the deprivation indices alone is limited because the level of deprivation is very likely established before the pandemic, it does not provide a state-dependent estimate of stress level in people who have encountered the pandemic. It would be helpful to examine the association with psychological/distress measures, such as anxiety and depression symptoms and social isolation indices (from available questionnaires) and potentially the interaction with deprivation indices.

Thank you for the suggestion. We believe a different dataset, study design, and future work is needed to explore mental health effects. Firstly, our study intentionally excluded individuals with depression and other major chronic health conditions (see Comment 8, from Reviewer 1 for further clarifications). This choice makes less likely the observed effects to be driven by pre-existing mental health issues. Secondly, the majority of anxiety and depression fields (e.g., Categories 138 and 140) were measured in 2017, years before the pandemic (only some self-reported fields are available longitudinally – see below). Consequently, they offer limited insights into pandemic-specific psychological effects. Thirdly, all groups in our analyses in the paper were matched for household size. Since household size influences social contact and potential isolation, this adjustment indirectly accounts for some aspects of pandemic-related loneliness that the Reviewer refers to. We have added a paragraph summarising these aspects in the Discussion and acknowledging the limitation that the deprivation indices do not

provide state-dependent measures. This section reads as follows: “*Our study design deliberately excluded individuals with major mental health conditions, reducing the likelihood that pre-existing depression or anxiety influenced our findings. Additionally, most available mental health data in the UKBB dataset, were collected years before the pandemic, limiting their relevance to pandemic-specific effects. While we matched groups for household size—an indirect measure of social contact and isolation—our deprivation indices do not capture state-dependent psychological stressors. Future research with longitudinal mental health data is needed to better understand the interplay between deprivation, stress, and brain ageing.*”

Nevertheless, as an exploratory analysis, we examined any available mental health data fields in the UK Biobank that were longitudinally measured and provided some state-dependent measures, all based on self-reported information. We considered the 36 mental health measures from the second imaging visit (e.g., loneliness, anxiety, and miserableness – Table P1) and their potential interactions with deprivation indices and pandemic exposure. We found no significant associations with RBAG after multiple comparison correction. We also tested for interactions between these mental health indicators and deprivation indices, but again, no significant effects were identified. For the reasons above we don’t think these negative findings support valid inferences so chose to not include them into the revised manuscript.

Table P1. List of mental health data fields.

UKBB Data Field	Name
1920	Mood swings
1930	Miserableness
1940	Irritability
1950	Sensitivity / hurt feelings
1960	Fed-up feelings
1970	Nervous feelings
1980	Worrier / anxious feelings
1990	Tense / 'highly strung'
2000	Worry too long after embarrassment
2010	Suffer from 'nerves'
2020	Loneliness, isolation
2030	Guilty feelings
2040	Risk taking
2050	Frequency of depressed mood in last 2 weeks
2060	Frequency of unenthusiasm / disinterest in last 2 weeks
2070	Frequency of tenseness / restlessness in last 2 weeks
2080	Frequency of tiredness / lethargy in last 2 weeks
2090	Seen doctor (GP) for nerves, anxiety, tension or depression
2100	Seen a psychiatrist for nerves, anxiety, tension or depression
4526	Happiness
4537	Work/job satisfaction
4548	Health satisfaction
4559	Family relationship satisfaction
4570	Friendships satisfaction
4581	Financial situation satisfaction
4598	Ever depressed for a whole week
4609	Longest period of depression
4620	Number of depression episodes
4631	Ever unenthusiastic/disinterested for a whole week
4642	Ever manic/hyper for 2 days
4653	Ever highly irritable/argumentative for 2 days
5375	Longest period of unenthusiasm / disinterest
5386	Number of unenthusiastic/disinterested episodes
5663	Length of longest manic/irritable episode
5674	Severity of manic/irritable episodes

20) The high correlation between predicted brain ages of participants at the two time points in figure 1e are likely driven by age and cannot indicate high scan-rescan model reproducibility.

Thank you for this comment, which is very similar to R2-14. We initially reported Pearson correlation values, following similar studies, for instance (Zhao et al, *Nature Mental Health* 2024). However, we acknowledge chronological age could have an effect on the observed correlation between predicted brain ages at two time points. To address this concern, we conducted additional analyses:

i) **Intraclass Correlation Coefficient (ICC):** As suggested by R2, we calculated the ICC for brain age predictions across the two scans, providing a measure of scan-rescan reliability that focuses on the stability of each participant's brain age estimate over time. The high ICC values (e.g., 0.98) demonstrate strong scan-rescan consistency, largely independent of age effects.

ii) **Partial Correlation Controlling for Age:** We performed a partial correlation analysis, controlling for chronological age, to further evaluate the effect of age on the observed correlation. The results showed that, even after adjusting for age, a significant correlation (partial $r > 0.86$) remained between brain age predictions at both time points. This suggests that the observed correlation is not solely driven by age but also reflects the stability of brain age estimates over time.

For completeness, we have now included ICC and partial correlation analyses in the revised manuscript, both of which provide additional validation of our model's scan-rescan reproducibility. The Results section, under "*Performance of brain age prediction models*", as well as Figure 1e and its caption, have been updated to report both ICC and partial correlation results for clarity.

21) What is the cut-off for the high vs low deprivation score? Are these deprivation scores analysed from the baseline or follow-up?

We have previously included the cut-off criteria for high vs. low deprivation scores in the Suppl. Material. However, for clarity, we have now added this information to the main text within the Methods section under the "*Interaction Effects against Socio-Demographic Factors*" sub-section. Specifically, participants were categorised as 'high' or 'low' on socio-demographic factors using the following thresholds: those scoring above the 70th percentile were classified as 'high', while those scoring below the 30th percentile were classified as 'low'. These percentiles were derived both from the entire UK Biobank study population and separately based on the countries in which participants resided. Further details are available in the Supplementary Materials section titled "*Deprivation Indices*".

Regarding the second part of the query, the deprivation scores used in this study were derived from baseline data because these measures are the only available socio-economic indicators in the UK Biobank dataset. Specifically, there are no longitudinal deprivation scores available within the UK Biobank. The deprivation indices were collected in 2008, 2009, and 2010 for Wales, Scotland, and England, respectively, and follow-up measures for these indices do not exist. This is now clarified in the Suppl. Material sub-section section "*Deprivation Indices*".